# Plasmodium falciparum utilizes pyrophosphate to fuel an essential proton pump in the ring stage and the transition to trophozoite stage

**Omobukola Solebo[1], Liqin Ling[1,2], Ikechukwu Nwankwo[1], Jing Zhou[2], Tian-Min Fu[3,4], Hangjun Ke[1]\***

**1** Center for Molecular Parasitology, Department of Microbiology and Immunology, Drexel University College of Medicine, Philadelphia, Pennsylvania, United States of America, **2** Department of Laboratory Medicine, West China Hospital, Sichuan University, Chengdu, China, **3** Department of Biological Chemistry and Pharmacology, The Ohio State University College of Medicine, Columbus, Ohio, United States of America, **4** The Comprehensive Cancer Center, The Ohio State University, Columbus, Ohio, United States of America

\* hk84@drexel.edu

**Data Availability Statement:** All data generated in this study have been included in the main and supplementary figures.

## Abstract

During asexual growth and replication cycles inside red blood cells, the malaria parasite *Plasmodium falciparum* primarily relies on glycolysis for energy supply, as its single mitochondrion performs little or no oxidative phosphorylation. Post merozoite invasion of a host red blood cell, the ring stage lasts approximately 20 hours and was traditionally thought to be metabolically quiescent. However, recent studies have shown that the ring stage is active in several energy-costly processes, including gene transcription, protein translation, protein export, and movement inside the host cell. It has remained unclear whether a low glycolytic flux alone can meet the energy demand of the ring stage over a long period post invasion. Here, we demonstrate that the metabolic by-product pyrophosphate (PPi) is a critical energy source for the development of the ring stage and its transition to the trophozoite stage. During early phases of the asexual development, the parasite utilizes *Plasmodium falciparum* vacuolar pyrophosphatase 1 (PfVP1), an ancient pyrophosphate-driven proton pump, to export protons across the parasite plasma membrane. Conditional deletion of PfVP1 leads to a delayed ring stage that lasts nearly 48 hours and a complete blockage of the ring-to-trophozoite transition before the onset of parasite death. This developmental arrest can be partially rescued by an orthologous vacuolar pyrophosphatase from *Arabidopsis thaliana*, but not by the soluble pyrophosphatase from *Saccharomyces cerevisiae*, which lacks proton pumping activities. Since proton-pumping pyrophosphatases have been evolutionarily lost in human hosts, the essentiality of PfVP1 suggests its potential as an antimalarial drug target. A drug target of the ring stage is highly desired, as current antimalarials have limited efficacy against this stage.

**Funding:** This work was supported by a Career Transition Award from NIH/NIAID (K22AI127702) and an R21 grant from NIH/NIAID (1R21AI156735) to HK. The funders have no roles in study design, data collection, data analysis and manuscript preparation.

**Competing interests:** The authors declare no competing interests.

## Author summary

Membrane-bound proton pumping pyrophosphatases (H$^+$-PPases), also known as vacuolar pyrophosphatases (V-PPases), are single-subunit proton pumps powered by hydrolysis of pyrophosphate (PPi) rather than ATP. These ancient, ATP independent proton pumps have been evolutionarily conserved in bacteria, archaea, plants, and protists, but are absent in fungi and animals. While H$^+$-PPases were discovered in *Plasmodium spp*. 20 years ago, their significance in malaria parasites has remained unclear. In this study, we have unveiled the pivotal roles of PfVP1, *Plasmodium falciparum* vacuolar pyrophosphatase 1, in the early phases of the asexual developmental cycle, including the ring stage and the ring-to-trophozoite transition. Through multiple approaches, we have confirmed that PfVP1 is a PPi hydrolyzing proton pump in *P. falciparum*, indicating, for the first time, that PPi serves as an energy source in malaria parasites. Our results further indicate that PfVP1 is an excellent antimalarial drug target due to its essential physiological roles and its absence in the human hosts. Antimalarials targeting the ring stage are highly desired, as most drugs have limited efficacy against this metabolically less active stage.

## Introduction

Malaria is a threat to 40% of the world's population and claimed more than 619,000 lives in 2021 [1]. Among five *Plasmodium* species that cause human malaria, *P. falciparum* is the deadliest. In a human host, the malaria parasite grows exponentially in bloodstream RBCs, causing all clinical symptoms including death in severe cases. The 48 h Intraerythrocytic Development Cycle (IDC) can be divided into three major developmental stages, including the ring, the trophozoite, and the schizont. These stages typically take about ~ 20 h, ~ 18 h, and ~ 10 h, respectively. Within the RBC, the parasite resides in a vacuole and is surrounded by multiple membranes, including the parasite plasma membrane (PPM), the parasitophorous vacuolar membrane (PVM), and the red blood cell membrane (RBCM). A major task of the ring stage parasite is to export proteins to the host cell to increase its permeability and cytoadherence [2]. Over the ~ 20 h period, however, the parasite is not replicating DNA or expanding its biomass significantly. After the RBCM has been permeabilized by the Plasmodium Surface Anion Channel (PSAC) [3], or New Permeability Pathways (NPPs) [4], the trophozoite starts to grow rapidly, resulting in 16–32 progeny in the subsequent schizont stage.

It has been long recognized that the asexual stage parasites rely on glycolysis for ATP production [5,6]. Per one molecule of glucose consumed, the parasite makes 2 ATP and 2 lactate molecules, with a minimal percentage of glucose-derived carbons fed into the tricarboxylic acid cycle (TCA) [7]. Indeed, the parasite can tolerate deletions of many TCA cycle enzymes [7] and some components of the mitochondrial electron transport chain [8,9], implying that the mitochondrion is a negligible source of ATP in the blood stages. To overcome the energy constraint mediated by substrate-level phosphorylation, the trophozoite stage parasite runs a high rate of glycolysis and consumes glucose in a rate that is 100-times faster than normal RBCs [10]. Permeabilization of the RBCM in this stage also facilitates lactate disposal to avoid a metabolic blockage of glycolysis. With an intact RBCM, however, the ring stage is traditionally thought to be metabolically quiescent, with a low-level of glycolysis being sufficient to meet the energy demand of this parasite [11].

Recent studies, however, suggest that the ring stage parasite fulfills many energy-costly processes over the period post invasion [12–14]. Although the genome is not replicating at this stage, RNA transcription and protein translation are active to form a ring stage specific

proteome for all necessary activities [12]. The PTEX translocon catalyzes ATP hydrolysis to move hundreds of parasite proteins to the RBC cytosol and membrane throughout the ring stage [13]. Rather than being static, ring stage parasites undergo dynamic movement inside the RBC and display morphological changes between the classical ring and a deformable ameboid-like structure [14]. In addition, the ring stage parasite also needs to spend energy to pump protons across the parasite plasma membrane to maintain the plasma membrane potential ($\Delta\psi$) and cytosolic pH. It has been suggested that the ATP-consuming V-type ATPase is the major proton pump in trophozoite stage parasites [15]. However, no studies have shown how proton export is carried out in the ring stage. RNA-seq data suggest that subunits of V-type ATPase are not highly transcribed until the trophozoite stage [16] (**S1 Fig**). Thus, it remains unknown how the ring stage parasite pumps protons and meets its energy demand while running a low-level of glycolysis.

In this study, we discovered that the ATP independent, proton pumping pyrophosphatase PfVP1 (_Plasmodium falciparum vacuolar pyrophosphatase 1_), is the major proton pump during ring stage development. Proton-pumping pyrophosphatases, or $H^+$-PPases, catalyze the hydrolysis of inorganic pyrophosphate (PPi), a by-product of over 200 cellular reactions, while harnessing the energy to pump protons across a biological membrane [17]. $H^+$-PPase was first discovered in the plant tonoplast and was also named vacuolar pyrophosphatase [18]. While $H^+$-PPases are absent in fungi and metazoans, it has been evolutionarily conserved in bacteria, archaea, plants, and many protozoa [19]. In _Arabidopsis thaliana_, two types of $H^+$-PPases are present, AVP1 (type I) and AVP2 (type II), and they require different cations for optimal activity. AVP1 is potassium dependent and calcium insensitive whereas AVP2 is calcium dependent and potassium insensitive [20]. Interestingly, the _P. falciparum_ genome also encodes two types of $H^+$-PPases, PfVP1 (PF3D7_1456800) and PfVP2 (PF3D7_1235200) [21]. RNA-seq data suggest PfVP2 is barely transcribed [16], which is consistent with its non-essential role in the asexual stages [22]; by contrast, PfVP1 is highly transcribed throughout the IDC and exhibits a peak level in the ring stage [16] (**S1 Fig**). Our data reveal that the malaria parasite _P. falciparum_ employs PfVP1 to harness energy from pyrophosphate, an ancient energy source, to support vital biological processes in the ring stage when ATP supply is likely low.

## Results

### PfVP1 is mainly localized to the parasite plasma membrane (PPM)

Localization of the two vacuolar pyrophosphatases in _P. falciparum_, PfVP1 and PfVP2, remains unclear in literature as previous studies used polyclonal antibodies raised against the _Arabidopsis thaliana_ vacuolar pyrophosphatase 1 (AVP1) in wildtype parasites [23], which were unable to differentiate PfVP1 from PfVP2. We have previously knocked out PfVP2 and did not notice any growth defects in normal culture conditions [22]. Therefore, to study the localization and function of PfVP1 without any potential interference or compensation from PfVP2, we genetically tagged PfVP1 in the PfVP2 knockout (KO) parasite line [22]. Briefly, in the 3D7-PfVP2 KO line, we utilized the CRISPR/Cas9 system [24,25] to endogenously tag PfVP1 with either a triple hemagglutinin (3HA) tag or a monomeric fluorescent protein (mNeonGreen). Additionally, through gene editing of the endogenous copy, the tagged PfVP1 was placed under the control of the TetR-DOZI-aptamer system for conditional expression [26,27] (**S2 Fig**). Thus, two transgenic parasite lines were constructed, 3D7-PfVP2KO-PfVP1-3HA^apt and 3D7-PfVP2KO-PfVP1-mNeonGreen^apt. We also cloned 3D7-PfVP2KO-PfVP1-3HA^apt by limited dilution and obtained two pure parasite clones, B11 and G11, which were phenotypically indistinguishable (B11 was used for this study). The parasite lines were

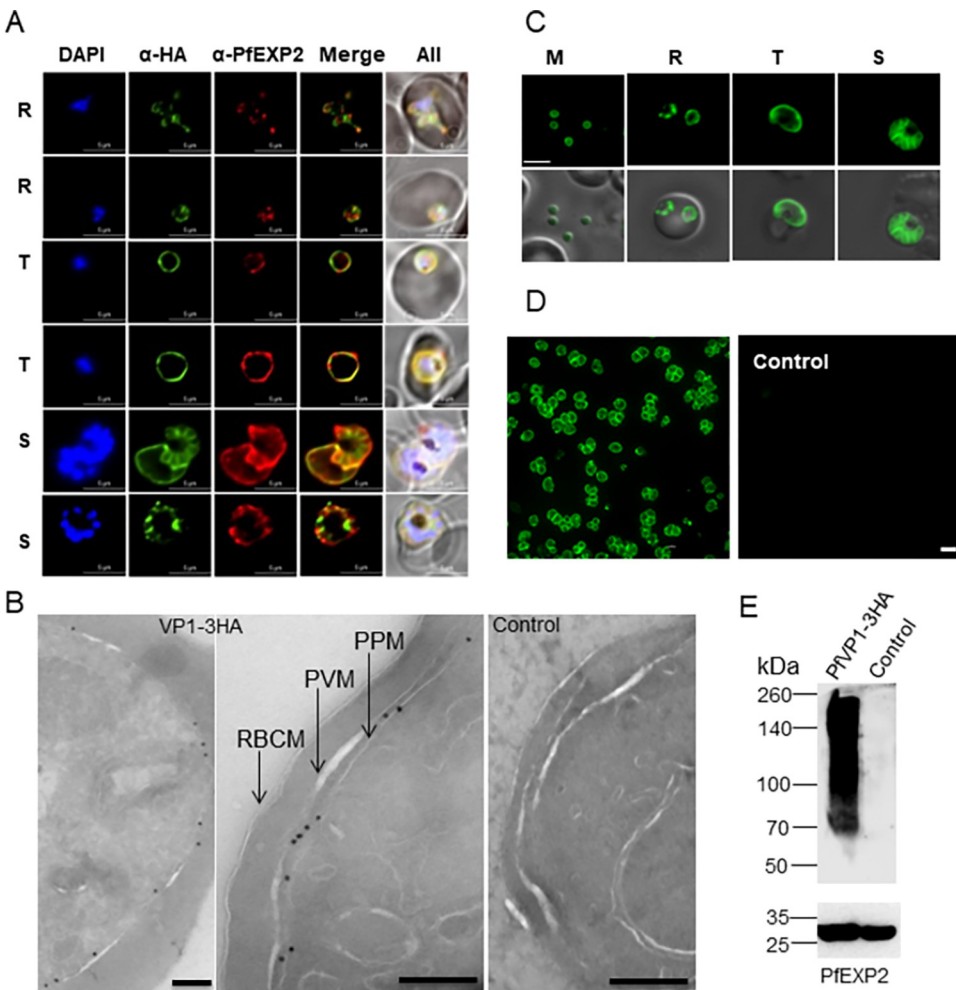

**Fig 1. PfVP1 is mainly localized to the parasite plasma membrane. A**, Immunofluorescence assay of 3D7-PfVP2KO-VP1-3HA[apt]. DAPI stains the nuclei. Green, PfVP1-3HA. Red, PfEXP2. Note an ameboid ring stage parasite in the first row. R, ring. T, trophozoite. S, schizont. Scale bars, 5 μm. Pearson Correlation Coefficient of green and red fluorescence was derived from n = 25 parasites of each stage: rings (0.5670±0.1401), trophozoites (0.9496 ±0.034) and schizonts (0.7508±0.197). **B**, Immunoelectron microscopy of 3D7-PfVP2KO-VP1-3HA[apt]. RBCM, RBC membrane. PVM, parasitophorous vacuolar membrane. PPM, parasite plasma membrane. Scale bars, 200 nm. 3D7 wildtype served as control. **C**, Live cell imaging of 3D7-PfVP2KO-VP1-mNeonGreen[apt]. M, merozoite. R, ring. T, trophozoite. S, schizont. Scale bar, 5 μm. **D**, Live cell imaging of Percoll enriched 3D7-PfVP2KO-VP1-mNeonGreen[apt] parasites. Scale bars, 10 μm. **E**, Western blot showing PfVP1-3HA's expression. The blot was re-probed with anti-PfExp2 antibody to show the loading control. Approximately, 15 μg of total protein lysate was loaded in the gel. 3D7 lysate served as a negative control.

normally cultured in the presence of 250 nM anhydrotetracycline (aTc) to maintain PfVP1 expression.

The subcellular localization of PfVP1 was verified by three methodologies, including immunofluorescence assay (IFA), immuno-electron microscopy (immuno-EM), and live cell microscopy (**Fig 1**). In the 3D7-PfVP2KO-PfVP1-3HA[apt] line, we performed colocalization studies using antibodies against HA and PfEXP2 [28]. PfEXP2 is a marker for the parasitophorous vacuolar membrane (PVM) [13], which served as an indicator of the parasite periphery. In the trophozoite stage, IFA revealed that PfVP1 was closely related to the PVM as it appeared to be co-localized with PfEXP2 (Pearson Correlation Coefficient 0.9496±0.034 from

25 parasites). In the ring and schizont stages, however, PfVP1's localization was distinct from that of PfEXP2 (Pearson Correlation Coefficient 0.5670±0.1401 for 25 rings and 0.7508±0.197 for 25 schizonts) (**Fig 1A**). Especially in the late schizont stage, the PPM invaginations were positive for PfVP1, not for PfEXP2, indicating PfVP1's localization on the PPM. To further verify this in a higher resolution, we performed immuno-EM studies and the data showed that PfVP1 was mainly localized to the PPM, not the PVM (**Fig 1B**). Quantification of 65 random images revealed 90% of the gold particles were localized to the PPM, ~ 2% of the gold particles were localized to nucleus/ER, and ~ 8% of the gold particles were localized to the cytosol or cytosolic small membranous structures with unknown identities (**S3 Fig**). Interestingly, no gold particles were apparently localized on the food vacuole, an organelle known to be acidified by proton pumps. To further verify this, we performed co-localization studies of PfVP1 and the food vacuole using the food vacuole marker, PfPlasmepsin II [29]. We were unable to find any parasites in which PfVP1 and PfPlasmepsin II were colocalized (Pearson Correlation Coefficient 0.6303±0.038 from 25 parasites) (**S4 Fig**). Thus, PfVP1 did not coincide with the food vacuole, in agreement with the previous publication [30]. Finally, we used live cell microscopy to verify PfVP1's localization in the 3D7-PfVP2KO-PfVP1-mNeonGreen[apt] line. In every stage of the IDC, including the merozoite, the ring, the trophozoite and the schizont, we detected strong fluorescence of PfVP1-mNeonGreen at the parasite periphery (**Fig 1C and 1D**). Since merozoites are free of PVM, the peripheral distribution of PfVP1 is consistent with its localization on the PPM, not the PVM. Together, we utilized multiple methods to show that PfVP1 is primarily localized on the parasite plasma membrane.

In the 3D7-PfVP2KO-PfVP1-3HA[apt] line, we also revealed that PfVP1 was highly expressed by Western blot (**Fig 1E**). We detected strong signals of PfVP1 in the transgenic parasite lysate, which were absent in the control lysate. Despite being specific, the band pattern of PfVP1 was highly unusual, ranging from the monomeric form of approximately 79 kDa to large, aggregated forms with molecular weights close to 260 kDa. This smearing band pattern of PfVP1 indicated that PfVP1 proteins were not completely solubilized by 2% SDS treatment. While the exact reasons for this phenomenon remain unclear at present, we speculate two possible explanations. One possibility is that the extremely hydrophobic nature of PfVP1, which possesses 16 transmembrane helices, may lead to solubility issues even in the presence of SDS. The other possibility is that PfVP1 works as oligomers in the parasites. Structural studies have shown that the orthologous VP1 protein in mung beans forms a homodimer to hydrolyze the substrate and pump protons [17]. It is plausible that PfVP1 works as dimers or other oligomeric structures in *P. falciparum*, which are not entirely solubilized by 2% SDS. This intriguing characteristic of PfVP1 warrants further investigation.

## Characterizing PfVP1 using the *Saccharomyces cerevisiae* heterologous system

To confirm PfVP1 is a PPi-dependent proton pump, we expressed PfVP1 in *Saccharomyces cerevisiae*. Since the 1990s, this heterologous system [31] has been widely applied to study many VP1 orthologs from plants and Archaea [32–34]. *S. cerevisiae* does not have VP1 homologs and thus provides a robust and clean system to study exogenous VP1 proteins [31]. Importantly, isolated yeast vacuolar vesicles incorporating recombinant VP1 are suitable for testing the pump's ability to move protons from one side of the membrane to the other. The yeast vesicles can also be used to examine VP1's enzymatic activities.

We transformed the yeast strain BJ5459 [35,36], which was null of the two major vacuolar proteases, PrA and PrB, with plasmids containing a copy of synthetic codon optimized PfVP1 (**S5 Fig**), AVP1 (*Arabidopsis thaliana* vacuolar pyrophosphatase 1), or a blank control. VP1

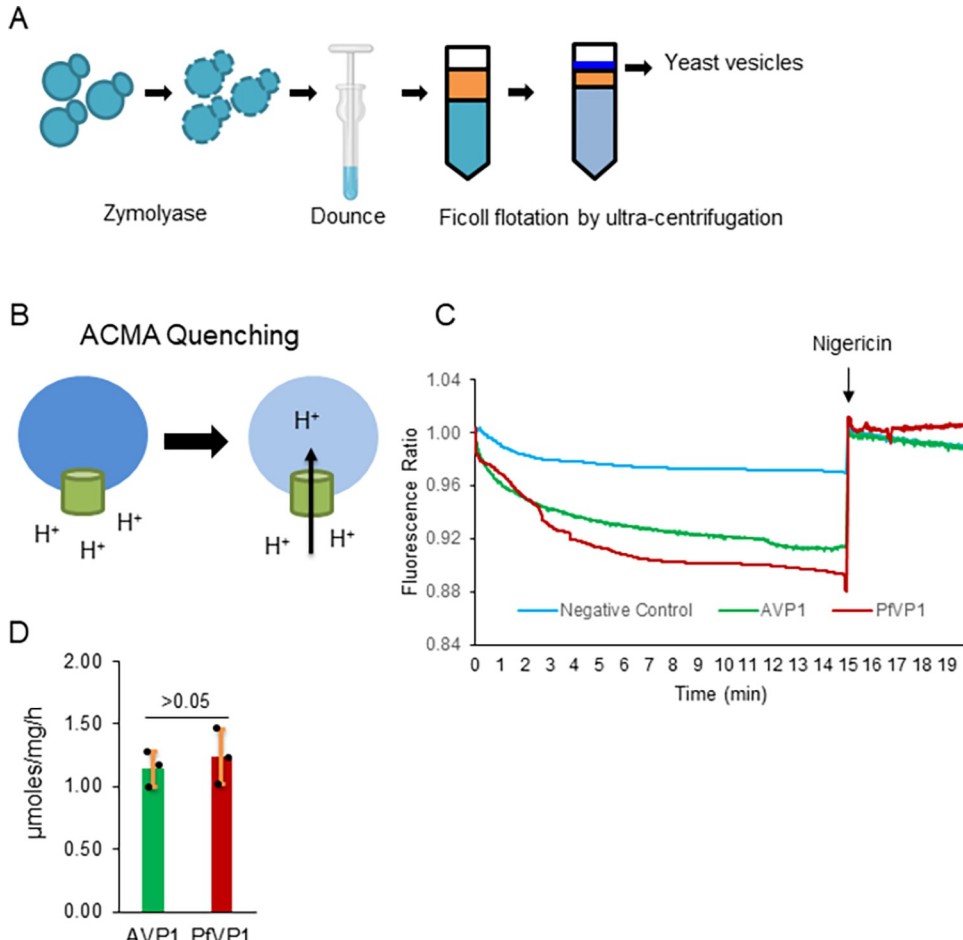

**Fig 2. PfVP1 is a PPi hydrolyzing proton pump. A**, A general schematic of purifying yeast vesicles bearing VP1 proteins from *Saccharomyces cerevisiae*. Yeast cells were treated with Zymolyase to remove the cell wall, lysed by Dounce, and applied to a Ficoll gradient (16% and 8%). After ultracentrifugation, yeast vesicles were collected from the top. This schematic was created with Biorender.com. **B**, A schematic of ACMA quenching experiment. **C**, ACMA quenching assay in isolated yeast vesicles bearing AVP1(*A. thaliana* vacuolar pyrophosphatase 1), PfVP1, or no VP1 protein (negative control). The ACMA's fluorescence signal was recorded after the yeast vesicles were added with the substrate, Na$_2$PPi, at time zero. Data shown are the representative of five individual experiments. **D**, Pyrophosphatase activity measurement in isolated yeast vesicles bearing AVP1, PfVP1, or no VP1 protein. The background activity from the negative control vesicles was subtracted from all measurements. This experiment was repeated three times with technical replicates. Statistical analysis was done by Student t-test, $p > 0.05$.

proteins were N-terminally tagged with the localization peptide of *Trypanosoma cruzi* VP1 (the first 28 amino acids) and GFP, which facilitates VP1's localization to yeast vacuoles [37]. The expression of PfVP1 and AVP1 in yeast was verified by fluorescence microscopy, which showed that the GFP signal appeared mainly on the yeast vacuoles (**S6 Fig**). Following the established protocols [38], we isolated yeast vesicles from three transformed lines expressing PfVP1, AVP1, or the blank control (**Fig 2A**). In the isolated yeast vesicles, a 9-Amino-6-Chloro-2-Methoxyacridine (ACMA) fluorescence quenching assay was used to assess the ability of VP1 to pump protons into the vesicle lumen (**Fig 2B**). The compound's fluorescence is quenched when a pH gradient forms across the vesicle membrane. The conditions of our proton pumping assays were set according to many published papers that studied VP1 proteins of protozoa and plant cells [21,23,38–44] (Materials and Methods). The yeast V-type ATPase, also present on the vesicles, was inhibited by Bafilomycin A1 to block its proton

pumping activity. Over time, the PfVP1 possessing vesicles were able to reduce ACMA fluorescence (**Fig 2C, red**). The positive control AVP1 expressing vesicles also quenched ACMA (**Fig 2C, green**), as expected, whereas the negative control vesicles bearing no H+-PPase exhibited little effect (**Fig 2C, cyan**). When Nigericin was added (a proton ionophore that abolishes transmembrane proton gradients), the quenched ACMA fluorescence was restored to its original levels (**Fig 2C**). This verified that the yeast vesicles were intact and PfVP1 and AVP1 expressing vesicles had accumulated protons inside. We also assessed PfVP1's enzymatic activity by measuring free Pi released by PPi hydrolysis (Materials and Methods). In comparison to the negative control vesicles (set to zero), PfVP1 expressing vesicles produced a net of 1.24 μmoles free Pi per mg of protein per h, similar to that produced by the AVP1 vesicles (1.14 μmoles/mg/h) (**Fig 2D**). Together, using the yeast heterologous expression system, we confirmed that PfVP1 is a PPi hydrolyzing proton pump. Our data also agree with the previously published results showing PfVP1's proton pumping and enzymatic activities in crude homogenates of malaria parasites [21,23,39].

## PfVP1 is essential for ring stage development and its transition to a trophozoite

To investigate PfVP1's essentiality during the 48 h IDC, we set up knockdown studies in highly synchronized cultures and examined parasite viability and morphology in the 3D7-PfVP2KO-PfVP1-3HA apt line. We used two approaches to remove aTc from cultures to initiate PfVP1 knockdown. In one approach, aTc was removed from Percoll isolated schizonts. In the other, aTc was removed from synchronized ring stage parasites. Both approaches led to similar knockdown phenotypes shown as follows.

When aTc was removed from schizonts, the knockdown culture did not display discernible defects in the 1st IDC (**Fig 3A**). This was likely because a prolonged time (~ 48 h) was needed to knock down > 95% of the PfVP1 protein (**S7A–S7B Fig**). In the 2nd IDC, the knockdown parasites successfully invaded new RBCs and established the ring stage. However, they showed minimal developmental changes from 72 h to 96 h (**Fig 3A**). It appeared that PfVP1 knockdown resulted in an extended ring stage as long as ~ 48 h (**Fig 3A**). Quantification of parasitemia and parasite morphological changes over the knockdown time course are shown in **Fig 3B and 3C**. The data showed that knockdown of PfVP1 caused severe developmental arrests from 72 to 96 h post aTc removal.

To better understand the morphological changes in the 2nd IDC, we further examined parasite development every 4 h (**Fig 3D**). Again, PfVP1 knockdown caused delayed ring stage development and a complete blockage of the ring to trophozoite transition. At the end of the 2nd IDC, we noticed that some knockdown parasites had expanded the cytosol slightly in comparison to parasites of earlier time-points and small hemozoin particles were also visible (**Fig 3D**), suggesting that the arrested parasites conducted some biochemical activities such as partial hemoglobin digestion. These results prompted us to verify whether the arrested parasites had established the New Permeability Pathway (NPP) [4]. This pathway is a result of the parasite's ability to insert a channel known as the Plasmodium surface anion channel (PSAC) to the RBC membrane [3]. The establishment of PSAC leads to the infected RBC releasing hemoglobin and undergoing osmotic lysis in response to high concentrations of small solutes. To investigate this, we treated the knockdown parasites at 72 h and 96 h post aTc removal with 500 mM alanine/10 mM HEPES and quantified the hemoglobin content in the supernatants. As shown in **S7C Fig**, at 96 h post knockdown, the aTc (-) culture exhibited a lower degree of osmotic lysis than that of the control. This observation suggests that the NPP was partially established in the knockdown parasites at 96 h after aTc removal. Despite this, however, none

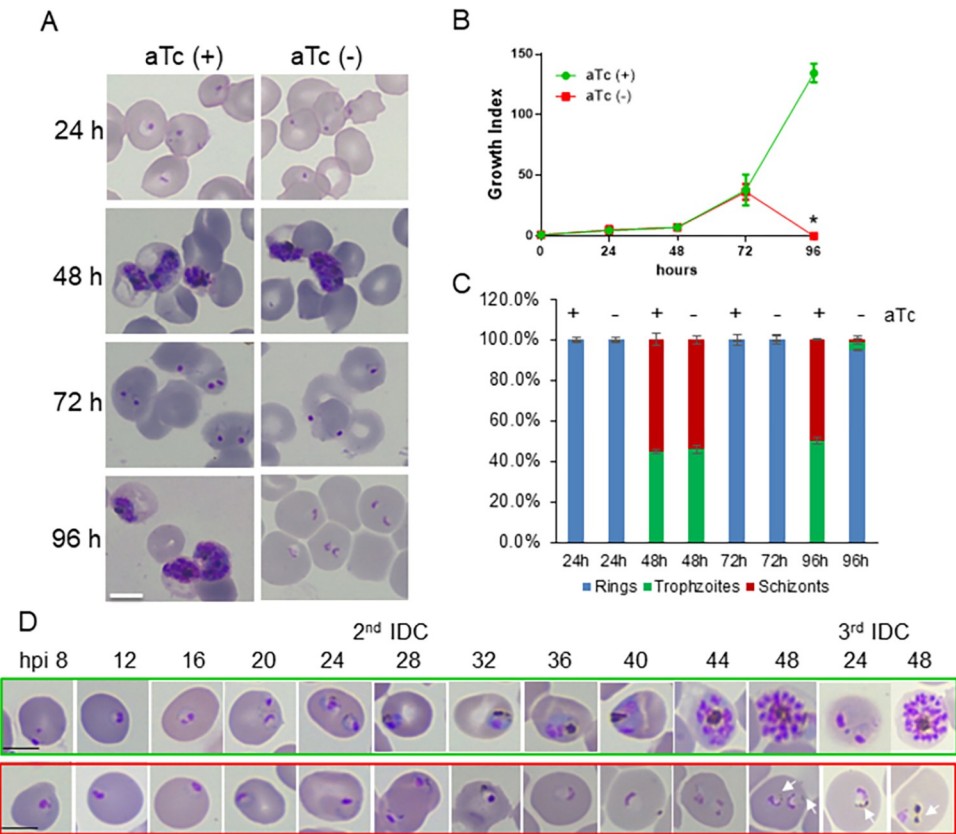

**Fig 3. PfVP1 is essential for the ring stage and its transition to trophozoite stage. A**, Knockdown experiment starting at the schizont stage by removal of aTc. Giemsa-stained thin blood smears of different time points are shown. Scale bar, 5 μm. This experiment was repeated more than 5 times. **B**, Parasitemia of the knockdown time course over 96 h. Parasitemia was determined by counting Giemsa-stained thin blood smears under a light microscope. Mean±s.d. of three replicates are shown. *, arrested parasites were not counted. **C**, Classification of parasite stages of the knockdown time course over 96 h. Mean±s.d. of three replicates are shown. In each thin blood smear, ~ 500 parasites were examined and classified into different morphological groups. **D**, Parasite morphological changes throughout the 2nd and 3rd IDC after aTc removal. Green box, aTc (+). Red box, aTc (-). Hours post invasion of each IDC are indicated. Scale bars, 5 μm. White arrows indicate small hemozoin particles in the knockdown parasites.

of the knockdown parasites showed the morphology of a normal looking trophozoite at this time. On the other hand, when the knockdown experiment was initiated from the ring stage, parasites also did not exhibit obvious defects in the 1st IDC (**S7D Fig**). However, in the 2nd IDC, the ring stage parasites failed to become trophozoites. Altogether, these results highlight that PfVP1 plays an essential role in ring stage development and its transition to the trophozoite stage.

To examine if the arrested parasites were viable, we added aTc back to the culture after aTc was removed for 96 h from schizonts. In agreement with **Fig 3D**, in the absence of aTc, the arrested ring stage parasites exhibited abnormal morphologies in the 3rd IDC and eventually lysed in the 4th IDC (**S8 Fig**). However, with aTc addback, the knockdown parasites were able to progress to mature stages and produced new ring stage parasites (**S8 Fig**). Quantification of parasitemia of the addback experiment indicated that aTc addback at 96 h post knockdown partially restored parasite growth and development (**S8 Fig**). Altogether, these data confirm that PfVP1 is essential for ring stage development and its transition to the trophozoite stage. Also, the ability of the knockdown parasites to recover after aTc addback indicates that *P. falciparum* has remarkable adaptation mechanisms to survive under stressed conditions.

## Phenotypic characterization of the PfVP1 knockdown parasites

We next characterized the knockdown phenotypes in the 3D7-PfVP2KO-PfVP1-3HA[apt] line. We reasoned if PfVP1 was genuinely a PPi-dependent proton pump located on the PPM, changes to cytosolic pH and PPi levels would be expected in the knockdown parasites. We used the ratiometric pH sensitive dye, 2',7'-Bis-(2-Carboxyethyl)-5-(and-6)-Carboxyfluorescein, Acetoxymethyl Ester (BCECF-AM), to measure cytosolic pH following the established protocol [15]. BCECF-AM is membrane permeable and is trapped inside the parasite cytosol after its ester group is removed. We initially aimed to measure cytosolic pH at 48, 72 and 96 h after aTc removal from schizonts. However, attempts to measure pH in ring stage parasites were unsuccessful. The failure was likely due to the small size or inadequate number of ring stage parasites per measurement. Nevertheless, our experience agrees with the fact that BCECF-AM has only been reported to measure cytosolic pH in trophozoite stage parasites [15,45,46], not in the ring stage.

At 48 h post aTc removal, we detected a small yet significant decrease of cytosolic pH in the knockdown parasites at the trophozoite stage (**Fig 4A**). This degree of pH drop from 7.41±0.05 to 7.17±0.11, however, did not negatively affect parasite growth and development as the knockdown parasites at 48 h post aTc removal exhibited no discernible abnormal morphologies (**Fig 3A**). To further test if such a small cytosolic pH change was specific to knockdown of PfVP1, we added aTc back to the knockdown culture on day 2 post aTc removal then measured pH again on day 4. After aTc addback for 2 days, the cytosolic pH of trophozoite stage parasites was restored to a normal level (7.35±0.07) (**Fig 4A**). This result suggested that the small, yet significant decrease of cytosolic pH was specific to PfVP1 knockdown. Previous studies have shown that a vacuolar pyrophosphatase specific inhibitor, IDP (imidodiphosphate), effectively inhibited pyrophosphate-dependent proton pumping in the crude parasite homogenates of *Plasmodium berghei* [39]. IDP apparently inhibited the VP1 protein in the rodent malaria parasites. To test if IDP could directly affect cytosolic pH in the human malaria parasites (*P. falciparum*), we added IDP (0.2 mM, the optimal concentration shown in [39]) to aTc plus and minus cultures on day 2 post knockdown and performed pH measurements. As shown in **Fig 4B**, IDP significantly reduced cytosolic pH to 7.18±0.03, in a degree similar to that of PfVP1 knockdown (7.17±0.11). Moreover, the combination of IDP and PfVP1 knockdown further reduced cytosolic pH to 7.05±0.01. Together, these data indicate that PfVP1 plays a role in maintaining cytosolic pH, supporting the mode of action of PfVP1 as a proton pump in *P. falciparum*.

Next, we measured PPi levels in the knockdown parasites using a newly developed PPi specific sensor (Materials and Methods). At 72, 84, and 96 h post aTc removal, saponin-lysed pellets were collected, and soluble metabolites were extracted using a mild process (Materials and Methods). In each sample, the concentrations of PPi and total parasite protein were measured, and the total amount of PPi (nanomoles) was normalized to total protein quantity (mg). We observed an increase of PPi at 84 h and 96 h post aTc removal (**Fig 4C**). Together, these data support PfVP1 works as a PPi driven proton pump in *P. falciparum*.

## Dual functionality of PfVP1 is critical for *P. falciparum*

All VP1 orthologous proteins have two functions, PPi hydrolysis and proton pumping. In some organisms, however, the two functions are not equally important. For example, in *A. thaliana*, the proton pumping activity of AVP1 is not essential, whereas the PPi hydrolysis activity is absolutely required. A previous study showed that a loss-of-function of AVP1 was rescued by expression the *S. cerevisiae* inorganic pyrophosphatase (yIPP1) [47]. The yIPP1 protein has the sole function of PPi removal, with no energy saving or proton pumping

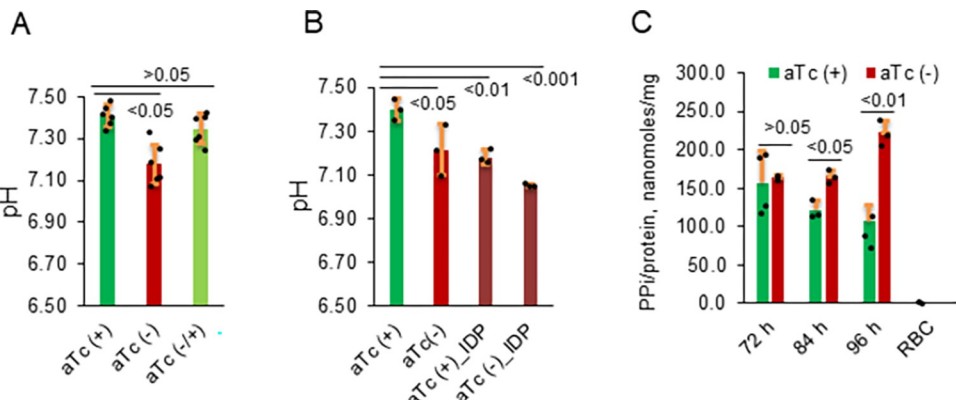

**Fig 4. Knockdown of PfVP1 alters the cytosolic pH and PPi levels. A**, In the 3D7-PfVP2KO-PfVP1-3HA[apt] line, pH was measured in the control culture (aTc, +), the knockdown culture after aTc removal for 48 h from the schizont stage (aTc, -), and the addback culture after aTc re-administration for 48 h from day 2 to day 4 (aTc, -/+). pH was measured using BCECF in a spectrofluorometer (Hitachi F-7000). Mean±s.d. of 6 measurements are shown. **B**, pH measurement of aTc plus and minus cultures challenged with IDP (imidodiphosphate). At 48 h post knockdown, IDP (0.2 mM) was added to the saponin treated parasites before pH measurement. Mean±s.d. of 3 measurements are shown. **C**, PPi measurement in the knockdown parasites after aTc removal for 72, 84, and 96 h from the schizont stage. Mean±s.d. of 3–4 measurements are shown. In A-C, statistical analysis was done by Student t-test. *P* values are shown in each panel. Experiments of A and B were repeated twice. Experiments of C were repeated three times.

activity. Therefore, in plant VP1 proteins, the proton pumping activity is less important than the PPi hydrolysis activity. To test if this scenario was also true for PfVP1, we performed a second round of transfection to complement the knockdown parasites with an episomal copy of yIPP1 (single function), AVP1 (dual function) or PfVP1 (dual function, control). The genes were cloned in plasmids containing hDHFR (human dihydroorotate dehydrogenase) [48] as the selectable marker. Since hDHFR was already used to knock out PfVP2 in the 3D7-PfVP2KO-PfVP1-3HA[apt] line, we made a new knockdown line in D10 wildtype, resulting in a D10-PfVP1-3HA[apt] line. This new PfVP1 knockdown line displayed the same phenotype as the 3D7 counterpart. We then transfected it with plasmids bearing 3Myc-tagged yIPP1, AVP1 or PfVP1 and obtained three individual transgenic lines (Materials and Methods).

Western blots showed that all Myc tagged copies were expressed independent of aTc, while the endogenous HA tagged PfVP1 was knocked down when aTc was removed for 96 h (**Fig 5A and 5B**). The raw Western blot images for Fig 5A and 5B were provided in **S9 Fig**. In **S9A Fig**, we once again observed the diffused band pattern of PfVP1 tagged with 3Myc in Western blot, which was consistent with the band pattern of PfVP1-3HA shown in **Fig 1E**. As mentioned earlier, we speculated that PfVP1 proteins were not well solubilized by 2% SDS, which resulted in this highly unusual band pattern. Notably, when a smaller amount of protein was loaded (3 μg in **S9D–S9E Fig** compared to 15 μg in **Fig 1E**), we could only detect the PfVP1 bands with larger molecular weights, but not the forms at monomeric sizes. This implied that the larger forms of PfVP1 predominated. Despite this unusual band pattern, C-terminal tagging of PfVP1 did not interfere with parasite physiology. This finding is consistent with other VP1 orthologs in Kinetoplastids [49] and *Toxoplasma gondii* [50], which also utilized C-terminal tagging. We next performed IFA to show that the complemented copies of PfVP1 and AVP1 were localized to the PPM whereas yIPP1 was in the cytosol (**Fig 5C**). When aTc was removed from synchronized D10-PfVP1-3HA[apt] schizonts for 96 h, the knockdown parasites were arrested in the late ring stage (**Fig 5D**), which is consistent with the knockdown phenotype of the 3D7 derived counterpart. In contrast, the knockdown parasites complemented with PfVP1-3Myc displayed normal growth like the aTc plus culture, indicating that the episomal

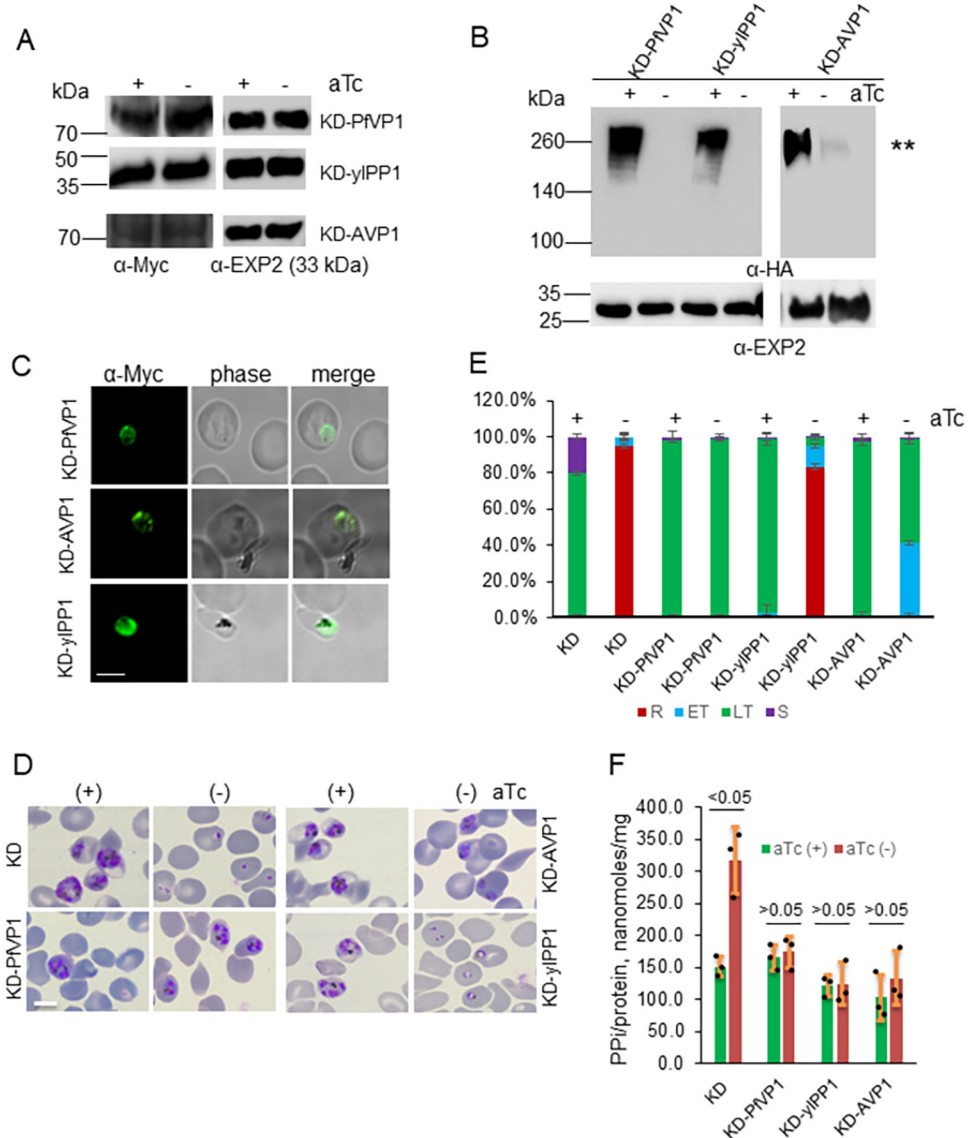

**Fig 5. The dual functionality of PfVP1 is required for parasite survival. A**, Western blot analysis of protein lysates from the D10-PfVP1-3HA[apt] line that was episomally transfected with Myc tagged PfVP1 (wildtype), yIPP1 (yeast inorganic pyrophosphatase), or AVP1 (*A. thaliana* vacuolar pyrophosphatase 1). Parasites were grown in aTc (+) or (-) conditions for 2 IDCs. Blots were probed with anti-Myc antibody and its secondary antibody. Duplicate samples were run and probed with anti-PfEXP2 to show loading controls. Approximately, ~ 30 μg of total protein lysate from each line was loaded in the gels. **B**, Western blot analysis of protein lysates from the D10-PfVP1-3HA[apt] line that was episomally transfected with Myc tagged PfVP1, yIPP1, or AVP1. Parasites were grown in aTc (+) or (-) conditions for 2 IDCs. Approximately, ~ 3 μg of total protein lysate from each line was loaded in the gels. Blots were first probed with anti-HA antibody to show the expression levels of the endogenously tagged PfVP1 with 3HA. **, when a small amount of total protein lysate was loaded, only aggregated forms of PfVP1 with high molecular weights were detected in Western blot. The same blots were re-probed with anti-PfExp2 antibody to show loading controls. **C**, Immunofluorescence assay (IFA). The complemented lines were probed with anti-Myc and a fluorescent secondary antibody. Scale bar, 5 μm. Representative images of n>30 parasites of each line were shown here. **D**, Morphologies of complemented parasite lines at 96 h after aTc removal. Giemsa-stained thin blood smears are shown. Scale bar, 5 μm. This experiment was repeated three times. **E**, Quantification of parasite morphologies in D. The percentage of different parasite morphological stages was determined by counting 1000 infected RBCs in each condition. R, ring. ET, early trophozoite. LT, late trophozoite. S, schizont. Mean±s.d. of three replicates are shown. This experiment was repeated two times. **F**, PPi measurement in the complemented parasite lines at 96 h post knockdown. Mean±s.d. of triplicate measurements are shown. This experiment was repeated four times. Statistical analysis was done by Student's t-test. *p* values are shown for each comparison. A-F, KD means knockdown.

PfVP1 fully rescued the loss of the endogenous VP1 protein (**Fig 5D**). AVP1 complementation displayed a moderate rescue with ~ two thirds of the parasites reaching the same morphology as control parasites (late trophozoites), and ~ one third progressing to a smaller size (early trophozoites). In contrast, yIPP1 was unable to restore parasite growth when the endogenous PfVP1 was knocked down. A quantification of various parasite morphologies in all conditions is shown in **Fig 5E**. To further verify if yIPP1 and AVP1 were enzymatically active in *P. falciparum*, we measured PPi levels in the aTc plus and minus cultures at 96 h post knockdown. As expected, knockdown of PfVP1 led to an elevated level of PPi in the parasites which was diminished in the lines complemented with PfVP1, yIPP1, or AVP1 (**Fig 5F**). Altogether, these data indicate 1) both PfVP1's PPi hydrolysis and proton pumping activities are essential for parasite survival, and 2) although not a 100% functional replacement, the orthologous plant VP1 is able to complement VP1-deficient *P. falciparum*, suggesting the functional conservation between PfVP1 and AVP1.

## Structure-guided mutagenesis studies of PfVP1 in *P. falciparum*

To further understand the mode of action of PfVP1, we conducted structure-guided mutagenesis studies in *P. falciparum*. All VP1 orthologs have 15–17 transmembrane helices with a molecular mass of 70–81 kDa [19]. The crystal structure of *Vigna radiata* (mung bean) VP1 (VrVP1) was resolved in 2012 [17]. At the primary sequence level, PfVP1 is highly similar to VrVP1 (49% identity and 66% similarity). The transmembrane (TM) helices are well conserved between PfVP1 and VrVP1, although the inter-domain loops display noticeable differences (**Fig 6A**). VrVP1 contains longer loops between the first three TMs. Based on the crystal structure, we computationally modeled the structure of PfVP1. The model showed a high degree of conservation of PfVP1 to VrVP1 with deviations in some loop regions (**Fig 6B**). The substrate binding and hydrolyzing sites of the modeled PfVP1 also mimic those of VrVP1 [17]. At the substrate binding site, all the conserved residues including 8 aspartates and 1 lysine are positioned around the substrate analog, the magnesium imidodiphosphate (MgIDP) (**Fig 6C**). The proton transfer pathway formed by TMs 5, 6, 11, 12 and 16 also appears to be structurally conserved (**Fig 6D**). These results indicate a high level of sequence and structural conservation between PfVP1 and the plant counterpart, VrVP1.

Based on these structural analyses, we chose to do alanine replacement mutagenesis of two putative substrate binding residues (D236, D461) and two residues that appear to be in the proton transfer channel and exit gate (D247, L697). Since *Plasmodium* is haploid and direct mutagenesis of essential residues would be lethal, we performed these mutagenesis studies in the D10-PfVP1-3HA<sup>apt</sup> line by episomal expression of mutated alleles (Materials and Methods). The effect of mutant PfVP1 on parasite viability was assessed upon knockdown of the endogenous copy by aTc removal. Fluorescence microscopy showed all mutant PfVP1 proteins were expressed and localized to the PPM (**Fig 7A**). When the endogenous HA tagged PfVP1 was knocked down by aTc removal for 96 h, all Myc tagged PfVP1 mutant alleles were still expressed (**Fig 7B**). The raw Western blot images for **Fig 7B** were provided in **S10 Fig**. In consistent with **S9 Fig**, we detected larger forms of PfVP1-3HA in a small amount of total protein lysate (3 μg) and observed PfVP1-3Myc migrated in the unusual smearing pattern. The mutant PfVP1 alleles had differing abilities to rescue the knockdown phenotype (**Fig 7C**). As a control, the episomal wildtype PfVP1 fully rescued the knockdown parasites grown in aTc minus medium. PfVP1 alleles with D236A and D247A mutations were unable to rescue the loss of endogenous VP1, indicating that these mutations abolished PfVP1's functionality. The D461A mutation had a low rescuing ability, but most D461A expressing parasites were unable to progress to the trophozoite stage. These results largely agreed with the results obtained with

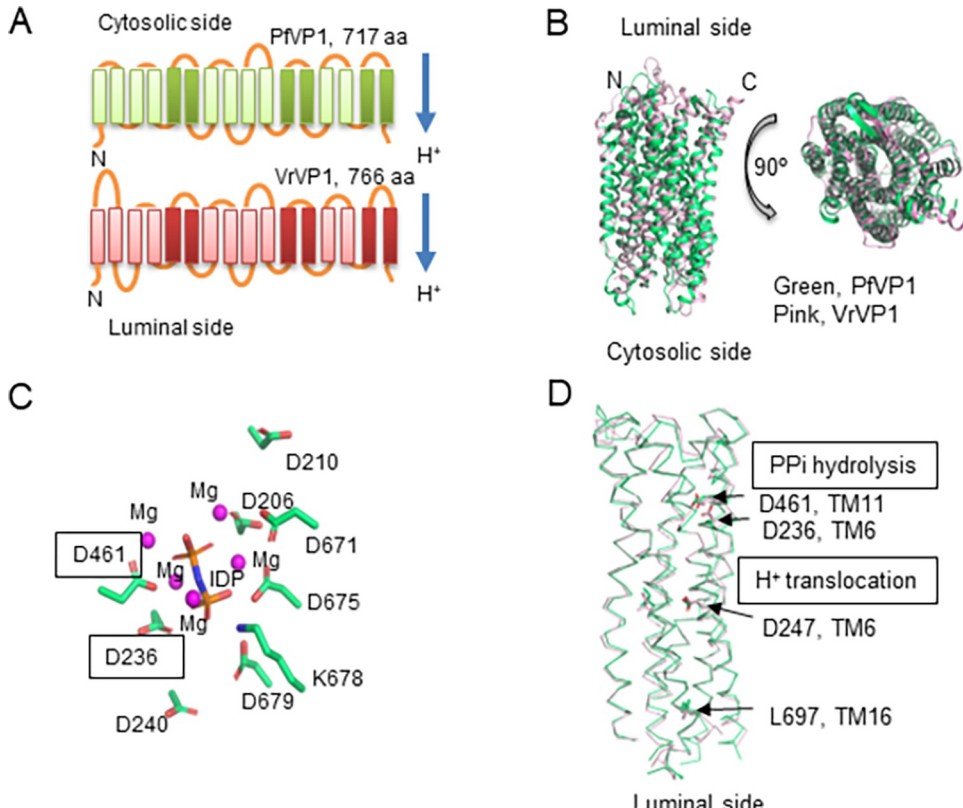

**Fig 6. The predicted 2D and 3D structures of PfVP1. A**, 2D schematic of PfVP1 and *Vigna radiate* VP1 (VrVP1) containing 16 transmembrane helices (TMs). In each monomer, TMs of 5, 6, 11, 12, 15, 16 (darker color) form the inner circle whereas the rest 10 TMs (lighter color) form the outer circle. Protons are pumped from the cytosolic side to the luminal side. **B**. Structure of PfVP1 (green) overlayed with the crystal structure of VrVP1 (pink). The PfVP1 structure was predicted using RoseTTaFold. **C**. Substrate binding site of PfVP1. The side chains of substrate binding amino acids were highlighted in sticks. Magnesium ions were shown in magenta spheres. IDP stands for imidodiphosphate, which was used to co-crystalize VrVP1[17]. Boxed residues will be mutated. **D**. Side view of the inner circle formed by TM5, TM6, TM11, TM12 and TM16. The proton transfer pathway is located at the lower part of the inner circle. Residues subjected to mutagenesis are indicated.

equivalent mutagenesis studies carried out in VrVP1 [43]. In contrast, PfVP1 differed from VrVP1 at the position of L697 (L749 for VrVP1). In mung beans, VrVP1 lost its proton pumping activity when L749 was mutated to alanine [43]. However, mutation of L697 to alanine in PfVP1 did not appear to cause any defects to parasite survival. A quantification of the rescuing ability of the various mutant PfVP1 alleles is shown in **Fig 7D**. Together, these results suggest that PfVP1 works as a PPi driven proton pump in *P. falciparum*. Although PfVP1 shares a high degree of structural similarity to VrVP1, it exhibits some variations from VrVP1 at positions such as near the proton exit gate.

We have listed the cell lines generated in this study and several key reagents in **Table 1**.

## Discussion

Our data indicate that PfVP1 is a critical proton pump in *Plasmodium falciparum* during ring stage development. PfVP1 is essential for *P. falciparum* to survive in the early phases of the asexual development, including the ring stage and the ring to trophozoite transition. PfVP1 is highly expressed and mainly localized to the parasite plasma membrane. Upon loss of PfVP1, the parasite undergoes a delayed ring stage development and a complete failure of

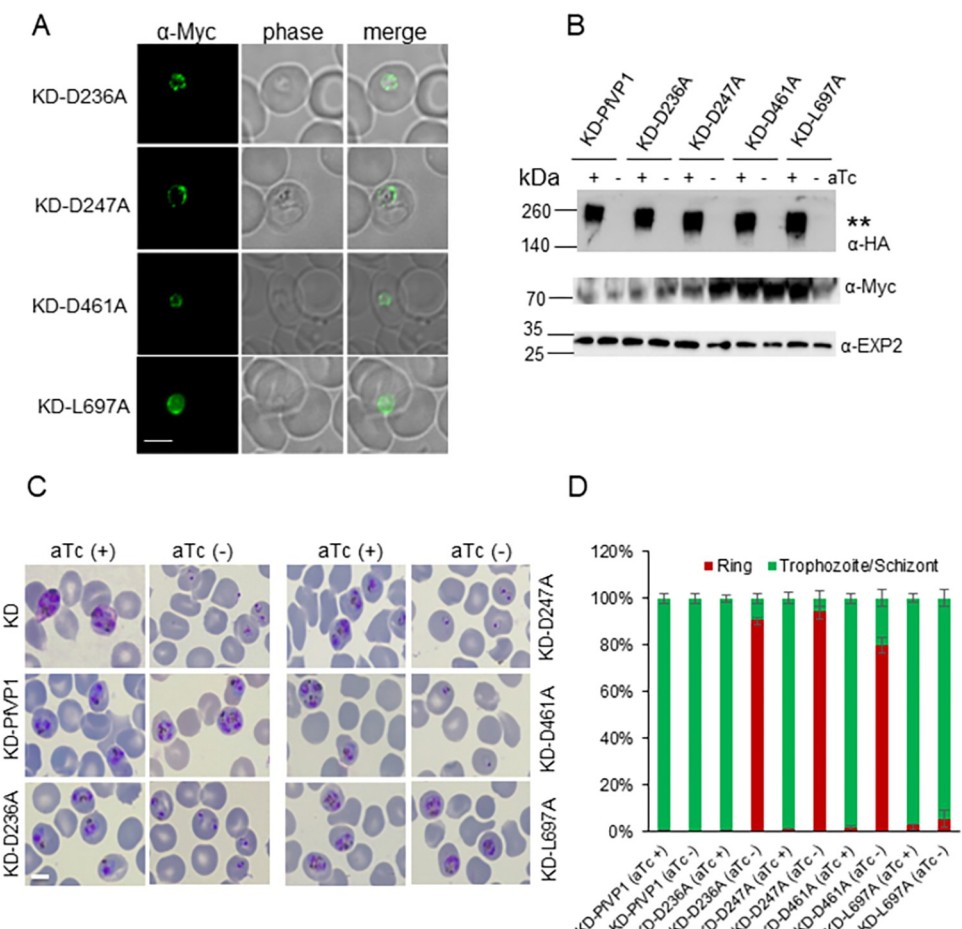

**Fig 7. Structure guided mutagenesis analysis of PfVP1. A**, Immunofluorescence assay (IFA) showing the localization of different mutated PfVP1 proteins tagged with 3Myc. Scale bar, 5 μm. Representative images of n>25 parasites of each line were shown here. **B**, Western blot analysis of protein lysates from various parasite lines expressing two copies of PfVP1, the endogenous PfVP1-3HA and the episomal PfVP1-3Myc. Parasites were grown in aTc +/- conditions for 96 h. Approximately, ~ 3 μg of total protein lysate from each line was loaded in the gels to detect the endogenous PfVP1-3HA. **, aggregated forms of PfVP1 with high molecular weights were detected in Western blot. The blot was re-probed with anti PfExp2 to show loading controls. A sperate gel was run to detect the episomally expressed PfVP1-3Myc (30 μg of total protein loaded per lane). **C**, Parasite morphologies of various PfVP1 lines at 96 h after aTc removal from the schizont stage. Giemsa-stained smears were shown. Scale bar, 5 μm. This experiment was repeated three times. **D**, Quantification of parasite morphologies in C. The percentage of different parasite morphological stages was determined from ~ 500 infected RBCs in each condition. Mean±s.d. of three replicates are shown. This experiment was repeated two times.

transitioning to the trophozoite stage. To our best knowledge, this is the first report showing that PPi is an energy source for *P. falciparum* to fulfill a critical biological function, at least in one part of the lifecycle.

While the IDCs of different malaria parasites vary between 24–72 h, the ring stage is invariably the longest period. In *P. falciparum*, the duration of the ring stage (~ 20 h) combined with the transition stage from the ring to trophozoite (~2–4 h) is about half of the entire IDC [14]. Inside the RBC, the ring stage parasite moves, changes its shape [14], and is actively exporting hundreds of proteins to the host cell [2]. Moreover, the ring stage is less susceptible to many antimalarial drugs and is the only stage that displays artemisinin resistance [51]. During the transition stage from the ring to trophozoite, the parasite also exhibits pronounced changes,

**Table 1. Key Resources.**

| Key Resources Table | | | | |
| --- | --- | --- | --- | --- |
| Reagent type | Designation | Source | Identifiers | Additional information |
| Cell line *P. falciparum* | 3D7-PfVP2KO | PMID: 21251930 | | Can be obtained from Vaidya lab |
| Cell line *P. falciparum* | 3D7-PfVP2KO-PfVP1-3HA[apt] | This study | | Can be obtained from Ke lab |
| Cell line *P. falciparum* | 3D7-PfVP2KO-PfVP1-mNeonGreen[apt] | This study | | Can be obtained from Ke lab |
| Cell line *P. falciparum* | D10-PfVP1-3HA[apt] | This study | | Can be obtained from Ke lab |
| Cell line *P. falciparum* | D10-PfVP1-3HA[apt]-PfVP1-3Myc | This study | | Can be obtained from Ke lab |
| Cell line *P. falciparum* | D10-PfVP1-3HA[apt]-yIPP1-3Myc | This study | | Can be obtained from Ke lab |
| Cell line *P. falciparum* | D10-PfVP1-3HA[apt]-AVP1-3Myc | This study | | Can be obtained from Ke lab |
| Cell line *P. falciparum* | D10-PfVP1-3HA[apt]-D236A(PfVP1)-3Myc | This study | | Can be obtained from Ke lab |
| Cell line *P. falciparum* | D10-PfVP1-3HA[apt]-D247A(PfVP1)-3Myc | This study | | Can be obtained from Ke lab |
| Cell line *P. falciparum* | D10-PfVP1-3HA[apt]-D461A(PfVP1)-3Myc | This study | | Can be obtained from Ke lab |
| Cell line *P. falciparum* | D10-PfVP1-3HA[apt]-L697A(PfVP1)-3Myc | This study | | Can be obtained from Ke lab |
| Cell line *S. cerevisiae* | BJ5459 | PMID: 32160104 | | |
| Cell line *S. cerevisiae* | BJ5459-AVP1 | This study | | Can be obtained from Ke lab |
| Cell line *S. cerevisiae* | BJ5459-PfVP1 | This study | | Can be obtained from Ke lab |
| antibody | anti-HA (Mouse monoclonal) | Santa Cruz Biotechnology | #sc-7392 | IFA (1:300) WB (1:10,000) |
| antibody | anti-Myc (Rabbit monoclonal) | Cell Signaling Technology | #2278 | IFA (1:300) WB (1:8,000) |
| antibody | anti-PfEXP2 (Rabbit polyclonal) | Burns JM Jr, Drexel University College of Medicine | | IFA (1:500) WB (1:10,000) |
| antibody | anti-Pf Plasmepsin II (Rabbit polyclonal) | Bei Resources | NIH/ NIAID | IFA (1:1000) |
| chemical compound, drug | IDP imidodiphosphate | Sigma | #I0631-1G | |

including a reduction in the parasite diameter, formation of several small hemozoin foci, and a transient echinocytosis of the host cell (RBC membrane distortion) [14]. Despite the significance of these early phases of parasite development, little is known about their cellular bioenergetics.

Earlier studies have shown that the ring stage parasite performs glycolysis at a much lower rate compared to that of the trophozoite stage [11]. Traditionally, a low-level of glycolysis is thought to be sufficient to support ring stage development. However, our study has revealed that the metabolic by-product PPi serves as a critical energy source during the early phases of the IDC. The free energy of PPi hydrolysis under physiological conditions is estimated to be -22.18 kJ/mol, which is a significant portion of the energy released from ATP hydrolysis (-37.6 kJ/mol) [52]. Evolutionarily, early life forms on earth used PPi as the energy source before

ATP emerged [52]. Apparently, the early divergent malaria parasite has evolutionarily reserved the ability to use PPi as a critical energy source, especially at the time when the ATP level is low. In the asexual blood stages, the malaria parasite primarily depends on glycolysis for ATP supply as the parasite's mitochondrion performs little or no oxidative phosphorylation. Therefore, PPi becomes a significant energy supplement in the ring stage parasite where glycolysis runs at a lower rate. Future studies will focus on quantifications of ATP and PPi throughout the IDC to understand their energetic contributions to *P. falciparum*.

What is the likely source of PPi in this metabolically less active ring stage parasite? PPi is a by-product of many synthetic reactions that use ATP (or other nucleoside triphosphate) to generate AMP (or other nucleoside monophosphate) and PPi, such as DNA, RNA, or protein synthesis. In the ring stage, PPi could be derived from a low level of RNA or protein synthesis since the parasite is not replicating DNA. Alternatively, PPi could be derived from some storage organelles such as acidocalcisomes, which store cations ($Ca^{2+}$, etc.) and polyphosphates [53]. It has been shown that *P. falciparum* merozoites contain acidocalcisomes [54]. The presence of acidocalcisomes could indicate that *P. falciparum* likely stores energy as polyphosphates in acidocalcisomes and liberates PPi from polyphosphates as needed when the ATP level is low. These intriguing aspects of acidocalcisome biology in *P. falciparum* deserve further investigations.

Unlike many other eukaryotes, malaria parasites generate the plasma membrane potential ($\Delta\psi$) through the transport of protons rather than sodium ions [55]. The proton gradient across the plasma membrane is also used by the parasite to perform secondary active transport to move ions, nutrients, or waste products into or out of the cell [55]. It has been long recognized that the *P. falciparum* possesses two different types of proton pumps, the single subunit PPi-dependent $H^+$-PPases [21,23] and the much faster ATP-dependent multi-subunit V-type ATPase [56]. Inhibition of the V-type ATPase by Bafilomycin A1 for 10–12 min causes a rapid drop of cytosolic pH from ~ 7.3 to ~ 6.8 in trophozoite stage parasites [15]. Therefore, the much slower proton pumps, PfVP1 and PfVP2, were thought to be insignificant or "marginal" to the parasite [56]. Alternatively, other studies have hypothesized that PfVP1 and/or PfVP2 would be critical to trophozoite stage parasites when energy demand is high [21]. In contrast to those earlier views, our results have now recognized the significance of PfVP1 for ring stage development and the transition to a trophozoite (PfVP2 is dispensable for asexual development [22]). We speculate that PfVP1's contribution to the early phases of development could be multifactorial, fulfilling tasks such as maintaining cytosolic pH, establishing plasma membrane potential ($\Delta\psi$), and redirecting ATP towards other energy-costly processes. Future studies are required to test these hypotheses. We acknowledge the challenges posed by working with ring stage parasites. For instance, the gold standard method of measuring $\Delta\psi$ using patch clamp electrophysiology has limited applicability to intracellular organisms like *Plasmodium* [57]. The small size of ring stage parasites further diminishes the feasibility of employing this technique. Clearly, alternative methodologies must be explored to thoroughly characterize ring stage biology in future.

It is interesting to note that PfVP1 is localized to a different subcellular compartment compared to most VP1 orthologs in other protozoa. As shown in this study, PfVP1 is primarily localized to the parasite plasma membrane. In contrast, the VP1 protein of *Trypanosoma brucei* [58] or *Trypanosoma cruzi* [59] is localized to acidocalcisomes and is used as the marker for these organelles. Interestingly, the orthologous VP1 protein in *Toxoplasma gondii* (TgVP1), another apicomplexan parasite related to *Plasmodium*, behaves drastically differently from the *Plasmodium* counterpart. TgVP1 is mainly localized to acidocalcisomes and the plant-like vacuole (PLV) [50,60] and despite phenotypic alterations, a complete knockout of TgVP1 is tolerated by the parasite [60]. In *P. falciparum*, however, the large-scale mutagenesis survey was

unable to disrupt the PfVP1 gene [61]. We have shown here that PfVP1 is essential for the ring stage development. Hence, the conserved VP1 proteins have seemingly adapted to perform different functions among different parasite linages even within the same phylum. It is likely that the unique bioenergetic features of *P. falciparum* necessitates PfVP1 to be localized on the parasite plasma membrane, performing proton pumping by hydrolyzing PPi, not ATP.

In summary, our data suggests that *P. falciparum* utilizes PfVP1 to harness energy from PPi to pump protons across the parasite plasma membrane. This process is crucial for parasite's development during the early phases of the asexual developmental cycle, including the ring stage and the transition from the ring to trophozoite. Additionally, the essential nature of PfVP1, coupled with its absence in humans, highlights it as a potential target for antimalarial drugs. Having a drug target in the ring stage is highly desirable for the drug development pipeline, as most antimalarials have limited efficacy against this metabolically less active stage. Indeed, efforts of developing inhibitors against $H^+$-PPases including PfVP1 have already begun [62].

## Materials and methods

### Plasmid construction

Detailed steps of plasmid construction are shown in the S1 Supplementary Information.

### Parasite culture, transfection, and knockdown studies

The 3D7-PfVP2KO (PfVP2 knockout) line was generated previously [22]. We used RPMI-1640 media supplemented with Albumax I (0.5%) to culture *P. falciparum* parasites in human $O^+$ RBCs as previously described [48,63]. We transfected *P. falciparum* ring stage parasites (~ 5% parasitemia) either with linearized or circular plasmid (~ 50 μg) using a BioRad electroporator. Post electroporation, parasite cultures were maintained in proper drug selections, e.g., blasticidin (2.5 μg/mL, InvivoGen), WR99210 (5 nM, a kind gift from Jacobs Pharmaceutical), and anhydrotetracycline (aTc) (250 nM, Fisher Scientific). Parasite synchronization was performed with several rounds of alanine/HEPES (0.5M/10 mM) treatment. For knockdown studies, synchronized parasites were washed thrice with 1xPBS to remove aTc and diluted in fresh blood (1:10) to receive aTc (+) or (-) media. Parasite cultures were Mycoplasma-free by routine PCR testing.

### Yeast culture, yeast lines and transformation

The *S. cerevisiae* strain BJ5459 was kindly supplied by Dr. Katrina Cooper from Rowan University [36], which was originally created by [35]. This strain (*MAT*a, *his3Δ200*, *can1*, *ura3–52*, *leu2Δ1*, *lys2–801*, *trp1-289*, *pep4Δ::HIS3*, *prb1Δ1.6R*) lacks yeast vacuolar proteases PrA (proteinase A) and PrB (proteinase B). Yeast cultures were maintained at 30°C either in YPD or Uracil drop-out medium. YPD medium contains 1% yeast extract (BP1422-500, Fisher Scientific), 2% peptone (20–260, Genesee Scientific), and 4% dextrose. Ura drop-out medium contains uracil minus complete supplement mixture (1004–010, Sunrise Science) and dropout base powder (1650–250, Sunrise Science). The latter has yeast nitrogen base (1.7 g/L), ammonium sulfate (5 g/L) and dextrose (20 g/L). Ura drop-out solid medium contains extra 2% agar. Yeast transformation was carried out using the Frozen-EZ Yeast Transformation II Kit (T2001, Zymo Research), according to manufacturer's protocols.

## Immunofluorescence analysis (IFA) and immuno-electron microscopy (Immuno-EM)

IFA was carried out as previously described [48,63]. Immuno-EM was performed at the Molecular Microbiology Imaging Facility at Washington University in St. Louis, MO. We used the following primary antibodies and dilutions: the HA probe (mouse, sc-7392, Santa Cruz Biotechnology; 1:300), the Myc probe (rabbit, 2278S, Cell signaling; 1:300), PfExp2 (rabbit, a kind gift from Dr. James Burns, Drexel University; 1:500), and PfPlasmepsin II (rabbit, Bei Resources, NIAID/NIH; 1:1000). We used fluorescently labeled secondary antibodies from Life Technologies (ThermoFisher Scientific) (anti-mouse or anti-rabbit, 1:300) or goat anti-mouse 18 nm colloidal gold-conjugated secondary antibody (Jackson ImmunoResearch Laboratories), as described previously [63]. Other details can be found [63].

## Alanine treatment and hemoglobin quantification

At 72 h and 96 h post aTc removal, aliquots of 100 μL aTc ± cultures (packed volumes) with 5% parasitemia were treated with 200 μL of alanine (0.5 M)/HEPES (10 mM) for 10 min at 37°C. The cell mixes were spun down at 2500 rpm for 2 min and the supernatants were transferred to a 96-well plate. The plate was subjected to OD measurement by Tecan infinite 200 pro at 405 nm. RBC was included as a negative control. In each condition, at least triplicate samples were measured.

## Western blot

Infected RBCs were lysed with 0.05% Saponin/PBS supplemented with 1x protease inhibitor cocktail (Apexbio Technology LLC) and protein was extracted with 2%SDS/62 mM Tris-HCl (pH 6.8) as previously described [63]. After protein transfer, the blot was stained with 0.1% Ponceau S/5% acetic acid for 5 min, de-stained by several PBS washes, and blocked with 5% non-fat milk/PBS. We used the following primary antibody dilutions: the HA probe (1:10,000), the Myc probe (1:8,000), and PfExp2 (1:10,000). We used HRP conjugated goat anti-mouse secondary antibody (A16078, ThermoFisher Scientific) at 1:10,000 or goat anti-rabbit HRP-conjugated secondary antibody (31460, ThermoFisher Scientific) at 1:10,000. Other steps followed the standard Bio-Rad Western protocols. For all Western samples, protein concentration was determined using the detergent tolerant Pierce BCA Protein Assay Kit (23227, ThermoFisher) according to the manufacturer's protocols. Blots were incubated with Pierce ECL substrates and developed by the ChemiDoc Imaging Systems (Bio-Rad).

## pH measurement using BCECF-AM (2',7'-Bis-(2-Carboxyethyl)-5-(and-6)-Carboxyfluorescein, Acetoxymethyl Ester)

We measured the pH of saponin permeabilized parasitized RBCs using the pH-sensitive fluorescent dye (BCECF-AM) according to published protocols [15]. For each measurement, 0.5-$1 \times 10^7$ parasitized RBCs were incubated with 4 μM BCECF-AM (B1170, ThermoFisher) and 0.02% Pluronic F-127 (p6867, ThermoFisher) in 16% hematocrit for 30 min at 37°C. Pluronic F-127 was used to facilitate the diffusion of BCECF-AM across cellular membranes. In both aTc (+) and aTc (-) conditions, one aliquot of parasitized RBCs was incubated with Pluronic F-127 alone to serve as a negative control. After incubation, infected RBCs were permeabilized with 0.05%Saponin/PBS and washed twice with warm saline/glucose buffer (NaCl 125 mM, KCl 5 mM, MgCl2 1 mM, glucose 20 mM, HEPES 25 mM, pH 7.4). The pellet was resuspended in 1 mL warm saline/glucose buffer, transferred to a cuvette, and placed in the temperature-controlled chamber of a spectrofluorometer (Hitachi F-7000). The cell suspension was

successively excited at 440 and 490 nm over 150 seconds and emitted fluorescence was measured at 535 nm. The ratio of fluorescence intensity excited by two wavelengths (490/440 nm) is a quantitative indicator of cellular pH.

To convert fluorescence intensity ratios to actual pH values, we calibrated pH measurement using the proton ionophore Nigericin according to published protocols [15]. Three aliquots of parasitized RBCs were incubated with BCECF-AM and Pluronic F-127 as described above, saponin treated, washed, and resuspended in a high $K^+$ saline buffer (KCl 130 mM, MgCl2 1 mM, glucose 20 mM, HEPES 25 mM) at a pH of 6.8, 7.1 and 7.8, respectively. Nigericin (20 μM, AAJ61349MA, Fisher Scientific) was added to each aliquot of cell suspension before the sample was placed in the spectrofluorometer. Emitted fluorescence was recorded at 535 nm by dual-wavelength excitation at 440/490 nm as described above. Linear regression of fluorescence intensity ratios and pH values yielded an equation (regression coefficiency >0.99), which was used to calculate pH values of individual samples.

### PPi extraction and measurement

PPi extraction was carried out following the published protocol with some modifications [64]. We used saline/glucose buffer (NaCl 125 mM, KCl 5 mM, MgCl2 1 mM, glucose 20 mM, HEPES 25 mM, pH 7.4) for saponin lysis and washes. At each timepoint from aTc (+) or (-) conditions, $2 \times 10^8$ parasitized RBCs (or uninfected RBCs as a control) were saponin lysed and washed 3 times to remove hemoglobin. The pellet was resuspended in 2–5 volumes of saline/glucose buffer, heated at 90˚C for 10 min to inactivate soluble pyrophosphatases, and saved at —80˚C. The samples were thawed and undergone 3 cycles of freezing/thawing between dry ice (10 min) and 37˚C (~ 2 min). They were sonicated for 30 min at 4˚C in a water bath sonicator (Fisher). After sonication, samples were spun down at 13,000 rpm for 10 min. The supernatants were saved for PPi measurement. The pellets were solubilized with 2%SDS/62 mM Tris-HCl (pH 6.8) overnight for protein quantification.

PPi was measured with a PPi fluorogenic sensor from Abcam (ab179836) (the chemical identity of this sensor was not released by the manufacturer). Briefly, 2 μL of each supernatant as extracted above was added into a 50 μL assay buffer containing 1:1000 diluted PPi fluorogenic sensor in a black plate. The mixture was incubated in the dark for 20–30 min and read by Tecan infinite 200 pro at 470 nm with excitation at 370 nm. A PPi standard curve was generated to determine PPi concentrations in samples.

### Yeast vesicle isolation

We followed published protocols to purify yeast vesicles expressing various VP1 proteins [38]. In brief, from the transformed plate, one colony was picked and inoculated into 5 mL of drop-out medium on Day 1. Day 2, the culture was diluted in 1:33 and grown overnight in 100 mL of drop-out medium. Day 3, the culture was diluted in 1 L of YPD (starting $OD_{600}$ = 0.05) and grown to reach $OD_{600}$ 0.8 (typically, < 12 h). The culture was then induced with 3 mM of $CuSO_4$ for 3–4 h to reach $OD_{600}$ between 1–1.3.

After induction, the yeast cells were pelleted, washed with deionized water and spheroplast buffer (1.2 M Sorbitol, 100 mM $KH_2PO_4$, pH 7.0), and weighed. The pellet was then resuspended in five volumes of spheroplast buffer containing 10 mM DTT and 1% glucose (both freshly added). In this mixture, per gram of wet yeast, 5 mL of Zymolyase 20T (120494–1, AMSBIO) at 5 mg/ml in 10 mM $Na_2HPO_4$, 50% Glycerol was added. This mixture was incubated at 30˚C while rotating for 2 h to digest the yeast cell wall. After digestion, the yeast pellet was washed twice with spheroplast buffer, resuspended in Lysis Buffer A (10 mM MES-Tris, 0.1 mM $MgCl_2$, pH 6.9, 12% Ficoll 400 (AAB2209518, Fisher Scientific, added fresh)). The

mixture was dounced 15–25 times on ice and pelleted. The supernatant was transferred to an ultracentrifuge tube and overlayed with layers of Lysis Buffer A with 12% Ficoll and Buffer B with 8% Ficoll (AAB2209518, Fisher) and centrifuged at 28,500 rpm for 45 min (Beckman 42.1). Afterwards, the wafer clump was collected using a pipette tip and resuspended in 3 mL of 2x Buffer C and mixed with 3 mL of 1x Buffer C (20 mM MES-Tris, 10 mM MgCl$_2$, 50 mM KCl, pH 6.9) and pelleted in the ultracentrifuge (Beckman 50Ti). The resulting pellet was then resuspended in 200 μL of 1x Buffer C with 10% glycerol, aliquoted into microcentrifuge tubes and flash frozen in an ethanol dry ice bath before storage at -80˚C.

### ACMA pH Quenching Assay

We measured proton pumping activities of VP1 in isolated yeast vesicles using the ACMA Fluorescence Quenching Assay [38]. ACMA stands for 9-amino-6-chloro-2-methoxyacridine (A1324, ThermoFisher Scientific). For each measurement in the spectrofluorometer (Hitachi F-7000), 30 μg of vesicles were added to the 1 mL of transport buffer (100 mM KCl, 50 mM NaCl, and 20 mM HEPES) in the presence of 1 μM of ACMA, 3 mM MgSO$_4$, 1 mM of Na$_2$PPi and 1 μM of Bafilomycin A1 (inhibitor of the yeast V-type ATPase). The reaction was monitored for 15 minutes to observe any decrease in fluorescence (excitation 410 nm, emission 490 nm). Afterwards, 10 μM of Nigericin is added to the solution and monitored for 4 min to see if the fluorescence could be restored.

### Pyrophosphatase activity measurement

The release of Pi by pyrophosphatase activity of VP1 proteins in isolated yeast vesicles was measured using the P$_i$Per $^{TM}$ Phosphatase Assay Kit (P22061, ThermoFisher Scientific), according to manufacturer's protocols. The Pi derived from PPi hydrolysis is coupled to three enzymatic reactions to convert a nonfluorescent compound (amplex red) to fluorescent resorufin. Potassium fluoride (KF, 0.25 mM) was added to inhibit the yeast soluble pyrophosphatase in the reactions. Fluorescence was detected by Tecan infinite 200 pro at 565 nm with excitation at 530 nm. A Pi standard curve was generated to determine Pi concentrations in the samples.

### Supporting information

**S1 Fig. RNA-seq data of pyrophosphatases and proton pumps in *P. falciparum*.** Transcription data was retrieved from PlasmoDB (deposited by Bartfai *et al.*[16]) and plotted. PfVP1, *Plasmodium falciparum* vacuolar pyrophosphatase 1; PfVP2, *Plasmodium falciparum* vacuolar pyrophosphatase 2; sPPase, *Plasmodium falciparum* soluble pyrophosphatase; V-type ATPase subunit B and subunit A. FPKM stands for fragments per kilobase of transcript per million mapped reads.
(TIFF)

**S2 Fig. Genetic tagging the endogenous locus of PfVP1 via CRISPR/Cas9. A**, Model of CRISPR/Cas9 mediated gene editing. The genetic locus of PfVP1 was tagged at the C-terminal with epitopes and aptamer repeats. Blasticidin deaminase served as the transfection marker. HR, homologous region. Green box, shield mutations within the gRNA coding region and 3HA. This schematic was created with Biorender.com. **B**, Model of aTc regulated conditional knockdown. Without aTc, the negative regulator, TetR-DOZI fusion protein, binds to the secondary structure of aptamers and protein translation is turned off. With aTc, the transcript is freed from TetR-DOZI and translated. aTc, anhydrotetracycline. TetR-DOZI, tetracycline repressor and development of zygote inhibited. **C**, Genotyping of 3D7-PfVP2KO-VP1-3HA$^{apt}$ by PCR. 5'int, 5' integration. 3'int, 3' integration. Star, a non-specific band. Primer positions

are shown in A.
(TIF)

**S3 Fig. Localization of PfVP1 via Immunoelectron Microscopy.** Representative immuno-EM images of 3D7-PfVP2KO-VP1-3HA<sup>apt</sup> labeled with anti-HA and gold-conjugated secondary antibodies. Black arrows indicate PVM, parasitophorous vacuolar membrane, and PPM, parasite plasma membrane. Red arrows indicate nuclear or cytosolic signals of PfVP1. Fv, food vacuole. Scale bars, 200 nm.
(TIF)

**S4 Fig. Co-localization of PfVP1 and the food vacuole via Immunofluorescence Assay.** DAPI stains nuclei. PfVP1 was detected by anti-HA and Alexa Fluor 488 anti-mouse secondary antibodies. The food vacuole was detected by anti-PfPlasmepsin II [29] and Alexa Fluor 568 anti-rabbit secondary antibodies. T, trophozoite. S, schizont. Representative images of n = 25 parasites of each stage are shown here. Pearson Correlation Coefficient (0.6303±0.038) of green and red fluorescence was derived from n = 25 parasites. Scale bar, 5 μm.
(TIF)

**S5 Fig. The sequence of codon optimized PfVP1 for expression in *S. cerevisiae*.** The start and stop codons of PfVP1 are highlighted in color. This synthetic DNA was made by a service company, Genewiz.
(TIF)

**S6 Fig. Localization of VP1 proteins in *S. cerevisiae*.** VP1 is N-terminally tagged with the localization peptide of TcVP1 (*Trypanosoma cruzi*) and GFP. Live cell microscopy showed localization of PfVP1 or AVP1 on the yeast's vacuole. The yeast cell transformed with a negative control plasmid showed a fuzzy background. Representative images of n = 30 yeast cells in each condition are shown here.
(TIF)

**S7 Fig. PfVP1 is highly expressed and essential for the ring stage and its transition to the trophozoite stage. A**, Western blot of 3D7-PfVP2KO-VP1-3HA<sup>apt</sup> parasites after aTc removal for 24 and 48 h from a highly synchronized culture. PfVP1 was detected by anti-HA and anti-mouse HRP conjugated secondary antibodies. PfExp2 served as a loading control. **, when a small amount of total protein lysate was loaded (~ 1 μg), only aggregated forms of PfVP1 with high molecular weights were detected by Western blot. **B**. The band intensity of the blots in A was quantified by ImageJ. **C**, The impact of alanine treatment on knockdown parasites at 72 h and 96 h post aTc removal. At each time point, parasite cultures were treated with 500 mM alanine in 10 mM HEPES and the supernatants containing hemoglobin were measured at OD405. Statistical analysis was done by Student t-test. **D**, Knockdown experiment starting at the ring stage by removal of aTc. Images were Giemsa-stained thin blood smears taken by a light microscope. White arrows indicate small hemozoin particles in the knockdown parasites. Scale bar, 5 μm.
(TIF)

**S8 Fig. Progression of the PfVP1 knockdown parasites after aTc addback. A**, Giemsa-stained images show parasite morphologies upon aTc addback after it was previously removed for 96 h. Scale bars, 5 μm. **B**, Parasitemia of the addback cultures. Parasitemia was determined by counting Giemsa-stained thin blood smears. Mean±s.d. of three replicates are shown. This experiment was repeated four times.
(TIF)

**S9 Fig. Raw Western blot images for Fig 5A and 5B. A**, Expression of the episomally complemented wildtype PfVP1 protein tagged with 3Myc (**A**), the yIPP1 protein tagged with 3Myc (**B**), and the AVP1 protein tagged with 3Myc (**C**). In A-C, 30 μg of protein lysate was loaded in each lane. Expression of the endogenous PfVP1 protein in the wildtype PfVP1 or yIPP1 complemented parasite lines (**D**) and in the AVP1 complemented parasite line (**E**). In D-E, 3 μg of protein lysate was loaded in each lane. Lanes in red boxes were cropped and shown in Fig 5A and 5B.
(TIF)

**S10 Fig. Raw Western blot images for Fig 7B. A**, Western blot analysis checking the endogenous PfVP1 protein in the PfVP1 knockdown parasites complemented with wildtype or mutant PfVP1 alleles. 3 μg of protein lysate was loaded in each lane. **B**, Western blot analysis checking the complementary PfVP1 protein in the PfVP1 knockdown parasites complemented with wildtype or mutant PfVP1 alleles. 30 μg of protein lysate was loaded in each lane. Lanes in red boxes were cropped and shown in Fig 7B.
(TIF)

**S1 Supplementary Information. Supplementary information contains information about plasmid construction.**
(DOCX)

## Acknowledgments

We thank former members of the Ke lab, Dr. Swati Dass, Neeta Shadija, and Dr. Maruthi Mulaka, for technical assistance. We thank members of Dr. Akhil Vaidya's lab at Drexel University for constructive discussions and Dr. Michael Mather, Dr. Ian Lamb, and Swaksha Rachuri for editing the manuscript. We thank Drs. Kendal Hirschi (Baylor College of Medicine) and Katrina Cooper (Rowan University) for providing yeast plasmids and strains. We thank Dr. James Burn (Drexel University), Dr. Daniel Goldberg (Washington University St Louis), and Dr. Joshua Beck (Iowa State University) for providing antibodies and plasmids. We thank Dr. Jacquin Niles (Massachusetts Institute of Technology) and Dr. Sean Prigge (Johns Hopkins University) for providing the knockdown tools. We thank Dr. Wandy Beatty (Washington University St Louis) for performing immune-EM studies.

## Author Contributions

**Conceptualization:** Hangjun Ke.

**Data curation:** Hangjun Ke.

**Formal analysis:** Omobukola Solebo, Liqin Ling, Ikechukwu Nwankwo, Tian-Min Fu, Hangjun Ke.

**Funding acquisition:** Hangjun Ke.

**Investigation:** Omobukola Solebo, Liqin Ling, Ikechukwu Nwankwo, Tian-Min Fu, Hangjun Ke.

**Methodology:** Omobukola Solebo, Liqin Ling, Ikechukwu Nwankwo, Tian-Min Fu, Hangjun Ke.

**Project administration:** Hangjun Ke.

**Resources:** Hangjun Ke.

**Supervision:** Jing Zhou, Hangjun Ke.

**Validation:** Hangjun Ke.

**Visualization:** Hangjun Ke.

**Writing – original draft:** Hangjun Ke.

**Writing – review & editing:** Omobukola Solebo, Liqin Ling, Ikechukwu Nwankwo, Jing Zhou, Tian-Min Fu, Hangjun Ke.

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
