## [Decision Letter · Decision Letter 0]

28 Jul 2023

Dear Dr. Ke,

Thank you very much for submitting your manuscript "Plasmodium falciparum utilizes pyrophosphate to fuel an essential proton pump in the ring stage and the transition to trophozoite stage" for consideration at PLOS Pathogens. As with all papers reviewed by the journal, your manuscript was reviewed by members of the editorial board and by several independent reviewers. In light of the reviews (below this email), we would like to invite the resubmission of a significantly-revised version that takes into account the reviewers' comments.

The reviewers were overall enthusiastic about the novelty and significance of this study characterizing a pyrophosphate-dependent proton pump and its essentiality for ring stage parasite development. As noted by the reviewers, the authors need to reassess the interpretations of their data and the strength of the conclusions that they draw from them. Additionally, the authors should resolve the issue raised by reviewer 2 concerning the variability of the signal ascribed to PfPV1 on Western blots in the manuscript. As suggested by the reviewer, images of full blots should be included as supplemental material.

We cannot make any decision about publication until we have seen the revised manuscript and your response to the reviewers' comments. Your revised manuscript is also likely to be sent to reviewers for further evaluation.

Sincerely,

Sean T Prigge

Guest Editor

PLOS Pathogens

Margaret Phillips

Section Editor

PLOS Pathogens

Kasturi Haldar

Editor-in-Chief

PLOS Pathogens

orcid.org/0000-0001-5065-158X

Michael Malim

Editor-in-Chief

PLOS Pathogens

orcid.org/0000-0002-7699-2064

The reviewers were overall enthusiastic about the novelty and significance of this study characterizing a pyrophosphate-dependent proton pump and its essentiality for ring stage parasite development. As noted by the reviewers, the authors need to reassess the interpretations of their data and the strength of the conclusions that they draw from them. Additionally, the authors should resolve the issue raised by reviewer 2 concerning the variability of the signal ascribed to PfPV1 on Western blots in the manuscript. As suggested by the reviewer, images of full blots should be included as supplemental material.

Reviewer's Responses to Questions

**Part I - Summary**

Reviewer #1: Solebo et al address the question of how the ring stage of P.f. parasite meets its energy demands and pump protons to maintain the plasma membrane potential. This is biochemically interesting because the mitochondrion is a negligible source of ATP in the blood stages and it is unclear whether glycolysis is sufficient for its energy demands. The authors hypothesized that an ATP-independent proton pump mechanism may be a major energy source during ring stages. In this study they focused on the proton pumping pyrophosphatase PfVP1 (Plasmodium falciparum vacuolar pyrophosphatase 1). Through endogenous tagging they determined its localization to the PPM and its essentiality to ring development. They used a yeast orthologous system to verify its enzymatic activity in proton pumping and PPi hydrolysis. They were able to show that PfVP1 knockdown results in a decrease in cytosolic pH and accumulation of PPi, leading to a reversible inhibition in ring to troph transition. Functional analysis through complementation of wild type and mutant enzymes further supported their hypothesis that PfVP1 works as a PPi driven proton pump. This is a well-designed study that address an interesting question through a nice combination of cellular, genetic and biochemical approaches. The writing is clear, the results convincing and include appropriate controls. I enjoy reading the manuscript and I think that it contributes significantly to our understating of parasite’s metabolism especially in the ring stages. I have mostly minor comments.

Reviewer #2: Solebo et al. carry out a series of carefully-conducted experiments on one of the pyrophosphate-utilising proton pumps present in the human malaria parasite P. falciparum. The authors show that the activity of this pump is essential for the ability of the parasite to transition from the ring stage to trophozoites. This is important new information for the field. I have one main issue pertaining to the experimental work (described below) and the authors have misinterpreted a relevant paper in the field. Both of these issues should be corrected before publication. In addition, some statements in the Discussion do not appear to be supported by the data presented and these should be removed or toned-down.

Reviewer #3: Clearly written, the concise new message of this paper is that the ancient and relatively low energy compound pyrophosphate, is a source of energy for P. falciparum plasma membrane proton pump pyrophosphatase PfVP1. Approaches used included study of pyrophosphatases of different organisms such as Arabidopsis Thaliana and Saccharomyces cerevisiae, and skillfully employed transfection of parasite with pyrophosphatase genes to elucidate the functions of PfVP1. The paper present new and important data about the functional role of PfPV1, the unique parasite specific drug target in the early stages of P. falciparum development.

One of the most important messages of this paper is that PfVP1, an essential protein, is critical for progression of intraerythrocytic cycle of parasite development, specifically in parasite transition from ring to trophozoite stage. Parasites depleted by PfVP1 by KD can still invade RBC, change shape inside the host cells, and digest hemoglobin but they are not able to visibly grow. One of the physiological markers of successful progression to the trophozoite stage of development is the establishing of the host cell membrane permeability (New Permeability pathway, NPP), which would be an accurate indicator of the time block inflicted by PfVP1 KO on parasite life. This information is missing, unfortunately, but data could be easily obtained in the simple sorbitol lysis assay. Certainly, the transcriptome profile would clarify the developmental time of the block in PfVP1-KD parasites, and it could give some insights into the block of parasite development, but this research may be beyond the task of the first report.

However, based on the presented results, the Reviewer is not certain, that the statements like “…PPi is the critical energy source for the ring stage…”, or “the delayed ring stage” are correct, because parasites in ring stage survive the absence of PPi hydrolysis in KO parasites, and parasites could be recovered from the block. It would be reasonable to make a more cautious statement about the need for PPi and functional PfVP1 in parasite growth, which is also the beginning of parasite development cycle. In the last example it would be more accurate to say that parasites are delayed in transitioning into the trophozoite stage. If the authors agree with this argument, then the text should be corrected in multiple places, starting from the title.

Minor deficiencies:

1. Text could be improved in several places by more accurate wording. Line 48: death is caused by P. falciparum infection but not by all human malaria parasite infections. Line 50: the length of the developmental stages is different between parasite strains. Line 52: parasite surrounded by two additional membranes – PVM and host PM. Page 60: missing words – per one molecule of glucose… Line 70: “immature parasite” is not an accurate term. Line 71: “Recent studies” need references. Line 72: “long period post invasion” is a murky definition. Line 117: missing word in “live cell microscopy” et cetera.

2. The description of PfVP1 localization is complicated with the analysis of co-localization of fluorescent signals from two proteins: PVM-embedded EXP2 and PfVP1. The close opposition of parasite and vacuolar membrane can be not be described quantitatively by fluorescent signals but requires electron microscopy analysis. The authors have, however, the sufficient fluorescent data indicating the PPM but not PVM has PfVP1: Fig 1 A shows the absence of the EXP2 in schizont PPM- invaginations decorated with PfVP1; Fig 1C clearly shows egressed merozoite with a strong VP1 fluorescence on the parasite periphery. This data is confirmed by immune-gold EM images. Authors do not have convincing evidence of other locations of VP1, and the word “primarily localized to the PPM” is not justified in the text (line 138 and in several other locations).

3. The accuracy of the statement in line 339 that “PfVP1 is the major proton pump in P. falciparum” likely needs to be rephrased. V-type ATPase is a more powerful proton pump controlling parasite pH (in trophozoites).

**Part II – Major Issues: Key Experiments Required for Acceptance**

Reviewer #1: (No Response)

Reviewer #2: Lines 140-149 – My main issue with the manuscript is the western blots – specifically the smearing observed, and the seemingly different sizes observed for PfVP1 in different western blots. The authors talk about a “regular amount”, but shouldn’t the amount be optimised for the protein of interest? It was not possible to see any distinct bands in the blot shown in Figure 1E. In fact, at the expected size for the protein (79 kDa) the smearing was the weakest. The authors indicate that this issue “warrants further investigation”, but I would argue that this needs to be sorted out for the present manuscript as western blots are used throughout the study with different size aggregates (depending on the amount of protein loaded and/or antibody used) being used to support their conclusions. In light of the smearing shown in the western in Figure 1E, I also think it’s inappropriate to then show highly cropped blots for most of the subsequent experiments. I suggest the authors include images of the full blots, if not in the main manuscript, then at least they should be included as supplementary information.

Line 134/135 – The sentence on these two lines is incorrect. The authors are encouraged to read reference 29 carefully (in particular the last paragraph of the paper), because we certainly did not say that PfVP1 is on the food vacuole. Quite the contrary, we stated that “the protein [PfVP1] was not present at detectable levels in isolated digestive vacuole preparations”. We then went on to say “This…. argues against PfVP1 being the digestive vacuole H+-PPase.” The data in the present manuscript therefore is consistent with the manuscript the authors state that their data contradicts!

Reviewer #3: see above

**Part III – Minor Issues: Editorial and Data Presentation Modifications**

Reviewer #1: 1. Line 7 page 3 (Introduction but also in abstract), red blood cells should be mentioned before using RBC for the first time.

2. Line 107 - Please justify the use of 3D7-PfVP2KO (knockout) parasite line as a background for endogenous PfVP1 tagging and cKD instead of wild type parasites (as appear later).

3. Figure S2 – For clarity, mark the endogenous gene in the upper illustration (PfVP1 (Pf3D7_1456800), WT) otherwise it might seem as if the insertion is on a null mutant background. 5HR is not very clear. Also, what is the green box? Why “new aptamers”. Please make the illustration more comprehensible for readers who are not well familiar with the tetR-Dozi system. Add numbers on primers in the illustration so readers can evaluate the integration test on C. Mark on A sizes in Kb that correspond to PCR fragments in C.

4. Figure 1 – How does the RNA-seq data from S1 (little to no expression in Schizont stage) correlate with the microscopy data (notable expression in schizonts)?

5. Lines 119-121 – Why would PfVP1 localize to the PVM during trophozoite stages?

6. Make sure title of S4 is the same in figure and in the legend in the WORD file.

7. In the attached Supplementary WORD file, legend for Figure S5 is missing and thus all subsequent legends are misnumbered and do not correspond to the main text and Sup figures numbering. Also, text in the legend does not match the text in the main article file, so it seems that this document needs to be edited and updated.

8. Line 161 - define AVP1.

9. S7 figure – Can’t see anything in the Ponceau staining so it’s hard to assess whether loading was comparable to conclude the efficiency of knockdown. Additionally, why does this blot look so differently than 1E? Where is the 70-80 kDa band that corresponds to the monomeric form? Why loading so little amount if the main form cannot be detected? Quantification of knockdown would be helpful.

10. Is the effect of IDP (imidodiphosphate) similar to PfVP1 knockdown? Does it block ring to troph transition? Couldn’t it block other, unrelated pyrophosphatases or proton pumps in P. falciparum? Does treating parasites with IDP phenocopies PfVP1 KD like in 3A, 4B or 4C? such data (which should be quite simple to obtain) would strengthen the claim that IDP acts on PfVP1 or at least blocks a similar mechanism. Please refer to the fact that IDP in its optimal concentration leads to a similar pH reduction as PfVP1 partial knockdown (before protein levels drop). A full block of PfVP1 by IDP should lead to failure in ring development already in the first cycle.

11. Figure 7 – how come the rescue construct (Myc-tagged) migrate at 70 kDa and not 260 KDa (aggregates??) like the endogenous copy?

Reviewer #2: Line 46-47 – The authors should ensure that they are using the most up to date numbers here.

Line 79 – Reference required for membrane potential.

Line 80 – Reference required for static pH level.

Line 97 – Is reference 21 correct here?

Lines 94/94 – The authors state as a fact the potassium and calcium dependence of PfVP1 and PfVP2. The authors need to clarify the basis for this statement and indicate whether or not this has been demonstrated experimentally.

Line 119 – The statement “Throughout the 48 h IDC, in the trophozoite stage,” needs to be fixed. The parasites are not in the trophozoite stage throughout the 48 h IDC.

Line 136 – The sentence that starts on this line is clumsy and should be amended.

Line 138 – Not sure of the utility of Figure 1D. It actually gives the impression of cytosolic localization. I suggest the authors delete this figure.

Line 200 – “are shown”

Line 208 – I do not believe that “meanwhile” is being used correctly here. Perhaps it should be “on the other hand”?

Line 214 – “In consistent” needs to be fixed.

Line 216 and elsewhere – I suggest the authors should refrain from using the term “addback” as this is jargon, presumably a term the authors used when discussing the experiments in-house.

Line 244 – I suggest the authors include a comment about the membrane permeability of IDP.

Line 297-298 – I think that the statement suggesting that, based on their data, VP1 is functionally conserved throughout evolution, is far too strong and should be deleted.

Line 339 – Seems very strange to start the Discussion section with “In summary”, especially because the last paragraph of the Discussion (line 415) also starts with “In summary”.

Line 339 – On what basis do the authors conclude that PfVP1 is “the major proton pump”? I’m not sure such a conclusion can be made. Yes, it is an important pump, but there could be another that is even more important (e.g. one that results in the death of ring-stage parasites (rather than their arrest) if it is functionally knocked out/down).

Line 383 – Should be reference 53.

Line 384 – I don’t think that reference 54 (a review) is appropriate reference here.

Line 396-397 – On what basis is the conclusion regarding the membrane potential made? I don’t believe the authors have data to support this statement.

Line 399 – Patch clamping of an intracellular organism has been achieved, but the ring stage is probably too small for the current setups. I believe this is the main problem.

The captions for the supplementary figures included in the supplementary file become out of order part way through because the caption for S5 is missing. I believe the same captions are correctly included in the main text.

Reviewer #3: see above

PLOS authors have the option to publish the peer review history of their article (what does this mean?). If published, this will include your full peer review and any attached files.

Reviewer #1: No

Reviewer #2: **Yes: **Kevin J Saliba

Reviewer #3: No
---

## [Decision Letter · Decision Letter 1]

31 Oct 2023

Dear Dr. Ke,

Thank you very much for submitting your manuscript "Plasmodium falciparum utilizes pyrophosphate to fuel an essential proton pump in the ring stage and the transition to trophozoite stage" for consideration at PLOS Pathogens. As with all papers reviewed by the journal, your manuscript was reviewed by members of the editorial board and by several independent reviewers. The reviewers appreciated the attention to an important topic. Based on the reviews, we are likely to accept this manuscript for publication, providing that you modify the manuscript according to the review recommendations.

The authors made appropriate efforts to respond to almost all of the issues raised by the reviewers during the first round of review. The remaining issue has to do with the behavior of VP1 on Western blots. The authors included uncropped images of relevant Western blots as supplementary data, but readers will likely still be interested in potential explanations of why VP1 appears as a smear on blots and sometimes appears to differ in molecular weight. As suggested in the current review, the authors should include in the manuscript text to acknowledge the behavior of VP1 on the blots and provide some discussion/explanation for this behavior.

Sincerely,

Sean T Prigge

Guest Editor

PLOS Pathogens

Margaret Phillips

Section Editor

PLOS Pathogens

Kasturi Haldar

Editor-in-Chief

PLOS Pathogens

orcid.org/0000-0001-5065-158X

Michael Malim

Editor-in-Chief

PLOS Pathogens

orcid.org/0000-0002-7699-2064

The authors made appropriate efforts to respond to almost all of the issues raised by the reviewers during the first round of review. The remaining issue has to do with the behavior of VP1 on Western blots. The authors included uncropped images of relevant Western blots as supplementary data, but readers will likely still be interested in potential explanations of why VP1 appears as a smear on blots and sometimes appears to differ in molecular weight. As suggested in the current review, the authors should include in the manuscript text to acknowledge the behavior of VP1 on the blots and provide some discussion/explanation for this behavior.

Reviewer Comments (if any, and for reference):

Reviewer's Responses to Questions

**Part I - Summary**

Reviewer #2: I thank the authors for making an effort to address the comments from all three reviewers. Whilst some of the concerns have been addressed and minor errors corrected, unfortunately the only issue I had with the data remains. The issue relates to the western blots. Reviewer #1 raised a similar concern, but only as a “minor issue” (their comment 11).

Although the authors did include the full blots in the supplementary section, the issue of the smearing was not addressed. Also not addressed was the fact that the protein appears to have difference sizes on different blots depending on how much is loaded. The author’s response to the different size issue does not make sense to me (the myc-tagged and HA-tagged copies of the protein do not appear to behave the same way on the blots). Having said this, I do believe that these issues do not impact on any of the major conclusions reached by the authors and it therefore becomes an editorial decision as to whether to accept the manuscript with the blots as they stand.

I would, however, suggest that the authors are asked to at least include some of their comments from the rebuttal into the actual manuscript so that readers of the manuscript do not have the same questions as the reviewers. An example of this is the authors’ response to comment 4 by reviewer #1, but there are a number of other similar instances that should be mentioned in the manuscript.

**Part II – Major Issues: Key Experiments Required for Acceptance**

Reviewer #2: (No Response)

**Part III – Minor Issues: Editorial and Data Presentation Modifications**

Reviewer #2: (No Response)

PLOS authors have the option to publish the peer review history of their article (what does this mean?). If published, this will include your full peer review and any attached files.

Reviewer #2: **Yes: **Kevin J Saliba

Figure Files:

Data Requirements:

Reproducibility:

References:

---

## [Editor Report · Decision Letter 2]

10 Nov 2023

Dear Dr. Ke,

We are pleased to inform you that your manuscript 'Plasmodium falciparum utilizes pyrophosphate to fuel an essential proton pump in the ring stage and the transition to trophozoite stage' has been provisionally accepted for publication in PLOS Pathogens.

Best regards,

Sean T Prigge

Guest Editor

PLOS Pathogens

Margaret Phillips

Section Editor

PLOS Pathogens

Kasturi Haldar

Editor-in-Chief

PLOS Pathogens

orcid.org/0000-0001-5065-158X

Michael Malim

Editor-in-Chief

PLOS Pathogens

orcid.org/0000-0002-7699-2064

The changes made in this revised version of the manuscript address the remaining issue raised by the reviewers. Please check line 381 in the manuscript where it says "In consistent with" when I think the authors mean "Consistent with".
---

## [Editor Report · Acceptance letter]

28 Nov 2023

Dear Dr. Ke,

We are delighted to inform you that your manuscript, "Plasmodium falciparum utilizes pyrophosphate to fuel an essential proton pump in the ring stage and the transition to trophozoite stage," has been formally accepted for publication in PLOS Pathogens.

Best regards,

Kasturi Haldar

Editor-in-Chief

PLOS Pathogens

orcid.org/0000-0001-5065-158X

Michael Malim

Editor-in-Chief

PLOS Pathogens

orcid.org/0000-0002-7699-2064